# A hypothalamic-thalamostriatal circuit that controls approach-avoidance conflict in rats

D. S. Engelke[1], X. O. Zhang [1], J. J. O'Malley[1], J. A. Fernandez-Leon [1], S. Li [2], G. J. Kirouac[2], M. Beierlein[1] & F. H. Do-Monte [1✉]

Survival depends on a balance between seeking rewards and avoiding potential threats, but the neural circuits that regulate this motivational conflict remain largely unknown. Using an approach-food vs. avoid-predator threat conflict test in rats, we identified a subpopulation of neurons in the anterior portion of the paraventricular thalamic nucleus (aPVT) which express corticotrophin-releasing factor (CRF) and are preferentially recruited during conflict. Inactivation of aPVT$^{CRF}$ neurons during conflict biases animal's response toward food, whereas activation of these cells recapitulates the food-seeking suppression observed during conflict. aPVT$^{CRF}$ neurons project densely to the nucleus accumbens (NAc), and activity in this pathway reduces food seeking and increases avoidance. In addition, we identified the ventromedial hypothalamus (VMH) as a critical input to aPVT$^{CRF}$ neurons, and demonstrated that VMH-aPVT neurons mediate defensive behaviors exclusively during conflict. Together, our findings describe a hypothalamic-thalamostriatal circuit that suppresses reward-seeking behavior under the competing demands of avoiding threats.

[1] Department of Neurobiology and Anatomy, The University of Texas Health Science Center, Houston, TX, USA. [2] Department of Oral Biol., University of Manitoba, Winnipeg, MB, Canada. ✉email: fabricio.h.domonte@uth.tmc.edu

I n nature, animals are constantly exposed to conflicting situations that involve both rewarding and risky components, challenging them to make decisions to maximize survival rate. For example, animals respond to predator threats with a series of defensive behaviors that decrease their chances of being attacked[1]. However, allocating the time and effort to avoid potential predators reduces foraging behavior and the probability of obtaining food in the environment[2]. In contrast, engaging in foraging behaviors renders the animal vulnerable to the risk of predation[3]. Therefore, survival demands a balance between seeking food (approach) and preventing possible harm (avoidance). While major advances have been made in elucidating the neural circuits underlying reward seeking and threat avoidance separately, the neural circuits that gate behavior amid approach-avoidance motivational conflict are not known.

The paraventricular nucleus of the thalamus (PVT) is a midline thalamic region that has received special attention in recent years due to its role in regulating diverse biological responses[4,5], including stress[6,7], fear[8,9], anxiety[10,11], and food seeking[12–15]. In rodents, the PVT receives dense projections from the prelimbic subregion of the prefrontal cortex (PL) and ventromedial hypothalamus (VMH)[16,17], two regions indispensable for food-seeking and anti-predator-threat responses, respectively[18–20]. In turn, the PVT is the major source of glutamatergic inputs to the nucleus accumbens (NAc)[21], a region implicated in the regulation of motivated behaviors[22,23]. These patterns of functional connectivity place the PVT in a unique position to control both reward and defensive responses during an approach-avoidance conflict. We hypothesized that PVT neurons integrate food-associated cues with predator-related threat signals to guide the most appropriate behavioral strategy. To test this hypothesis, we used an ethologically relevant experimental model in which rats are exposed to cues that predict a natural threat (cat odor) while foraging for reward (food). Using a combination of chemogenetics, optogenetics, and electrophysiological recordings in freely behaving rats, we identified a hypothalamic-thalamostriatal circuit that is activated by predator threat and is indispensable to regulate the opposing drives of approaching food and avoiding the potential risk of predation.

## Results
### Cat odor induces defensive behaviors and suppresses food seeking in rats. 
To characterize the behavioral conflict between approaching a source of food and avoiding predator threats, we established a model in which rats need to balance anti-predator defenses with food-seeking responses (Fig. 1a and Supplementary Movies 1 and 2). In this model, food-restricted rats are initially trained to press a lever for sucrose pellets during a period in which an audiovisual cue is presented to signal food availability (Supplementary Figure 1a)[12]. During the test day, a piece of filter paper soaked with cat saliva is positioned close to the food dish and is used to induce defensive responses (see Methods). Cat saliva has been shown to contain chemosignals that bind to the vomeronasal organ, activating the accessory olfactory bulb and other downstream regions to trigger a set of anti-predator defensive responses[24].

We found that when faced with both food cues and cat odor during the conflict test, rats displayed a repertoire of defensive behaviors including freezing, avoidance, and head-out (risk-assessment) responses (Fig. 1b–e). In addition, rats showed a strong suppression of food-seeking responses characterized by three distinct behavioral changes: (1) a reduction in the time spent exploring the food area, (2) an attenuation in the rate of lever presses, and (3) an increase in the latency to press after the onset of the food cue (Fig. 1f–h). The suppression in food-seeking responses persisted the following day when animals were returned to the same chamber in the absence of cat odor (Supplementary Figure 1b–h). While both male and female rats showed increased defensive behaviors and attenuated food-seeking responses during the conflict test, female rats had significantly lower rates of lever pressing (Supplementary Figure 2a–p). Because foraging depends on internally generated metabolic signals (e.g., hunger, caloric needs)[25], we asked whether longer periods of fasting could bias rat's behavior toward food seeking. Rats undergoing 72 h of food deprivation showed an attenuation in freezing and risk-assessment responses, when compared to 24 h food-deprived rats (Supplementary Figure 3a–d). However, both 24 and 72 h food-deprived rats displayed similar levels of food-seeking suppression during the conflict test (Supplementary Figure 3e–g). Together, these results demonstrate that our conflict test is a useful model to investigate the neural circuits underlying food-approach vs. predator-avoidance interactions.

### Exposure to cat odor activates the aPVT and other brain regions involved in anti-predator defenses. 
To begin identifying the brain regions involved in anti-predator defensive responses, we quantified the expression of the neural activity marker cFos in brain sections collected from rats exposed to either cat odor or neutral odor (Supplementary Figure 1i–m). Exposure to cat odor increased the number of cFos-positive neurons in several brain regions previously shown to be activated by predator odor[26], with a higher number of immunoreactive cells observed in the posteroventral division of the medial amygdala (MeApv), dorsomedial-central portion of the VMH (VMHdm-c), PL, and dorsomedial and dorsolateral subregions of the periaqueductal gray matter (PAGdm-dl) (Fig. 1j–n). Interestingly, exposure to cat odor also increased cFos expression in the anterior aspect of PVT (aPVT) (Fig. 1o), a region implicated in the control of food-seeking behaviors[12]. We, therefore, explored the potential role of aPVT in regulating the shift between avoiding predator threats and searching for food.

### aPVT neurons change their firing rates during the conflict. 
To record the activity of individual aPVT neurons, we performed extracellular recordings during different phases of the conflict test: (i) *food-seeking phase*, only food cues presented; (ii) *cat odor phase*, only cat odor presented, and (iii) *conflict phase*, food cues concomitantly presented with cat odor (Fig. 2a, b). These three phases were conducted during a single recording session to ensure that the same cells were tracked throughout the session and across phases. When we examined changes in spontaneous activity before vs. after each phase, we observed three different subsets of neurons that were exclusively responsive to (i) the food-seeking phase, (ii) the cat odor phase, or (iii) the conflict phase. In addition, we observed a different subset of neurons that responded nonselectively to more than one phase (Fig. 2c). Notably, the majority of the cells that were either excited or inhibited during the food-seeking phase responded in opposite direction during the cat odor phase, suggesting that most aPVT neurons change their firing rate according to the valence of the stimulus (Supplementary Figure 4h, i). Next, we examined the neuronal activity of aPVT neurons time-locked to the onset of the food cues in the absence of cat odor (before conflict), and compared it with the response of the same neurons in the presence of cat odor (during conflict). We identified two distinct subsets of neurons that either increased or reduced their firing rates in response to food cues (Fig. 2d). Remarkably, food-cue responses in both subsets of aPVT neurons were abolished during the conflict phase, and these neuronal changes were associated with

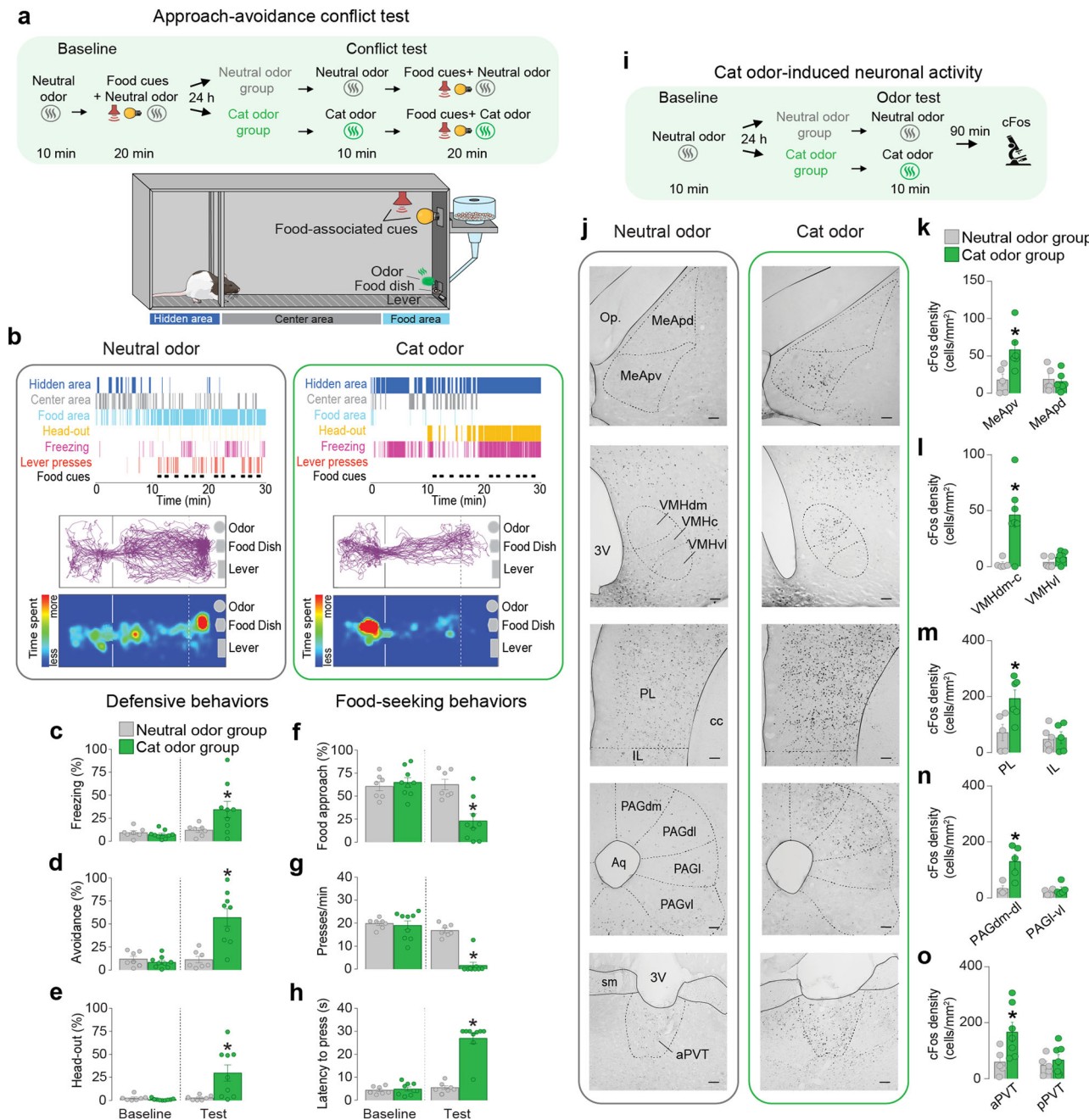

**Fig. 1 Cat odor exposure induces defensive behaviors, suppresses food seeking, and increases cFos expression in the aPVT. a** Timeline and schematic of the approach-avoidance conflict test. **b** Representative ethogram (top), tracks (center), and heatmaps (bottom) of rats exposed to neural odor (left) or cat odor (right). **c–h** During the conflict test, rats exposed to cat odor (green bars, $n = 9$) showed an increase in the percentage of time exhibiting freezing ($F_{(1, 14)} = 5.39$, $P = 0.035$), **d** avoidance ($F_{(1, 14)} = 21.85$, $P < 0.001$), and **e** head-out responses ($F_{(1, 14)} = 7.01$, $P = 0.019$); and a decrease in the percentage of time **f** approaching the food area ($F_{(1, 14)} = 28.31$, $P < 0.001$), a suppression in the number of **g** lever presses ($F_{(1, 14)} = 29.93$, $P < 0.001$), and a prolonged **h** latency to press the lever ($F_{(1, 14)} = 67.27$, $P < 0.001$), when compared to neutral odor controls (gray bars, $n = 7$; two-way repeated-measures ANOVA followed by Bonferroni post hoc test). **i** Timeline of the cat odor-induced neuronal activity test. **j** Representative micrographs of cFos immunoreactivity (dark dots) in neutral odor (left) and cat odor (right) groups. **k–o** Cat odor exposure (green bars, $n = 7$) increased the number of cFos-positive neurons in **k** the posteroventral subregion of the medial amygdala (MeApv, $P = 0.019$, $t = 2.834$), **l** the dorsomedial-central subregion of the ventromedial hypothalamus (VMHdm-c, $P = 0.0075$, $t = 3.34$), **m** the prelimbic cortex (PL, $P = 0.034$, $t = 2.44$), $n$ the dorsomedial and dorsolateral subregions of the periaqueductal gray matter (PAGdm-dl, $P = 0.015$, $t = 3.195$), as well as in the **o** anterior subregion of the paraventricular nucleus of the thalamus (aPVT, $P = 0.039$, $t = 2.372$), when compared to neutral odor controls (gray bars, $n = 5$, unpaired Student's $t$ test). pPVT posterior subregion of the paraventricular nucleus of the thalamus, MeApd posterodorsal subregion of the medial amygdala, VMHvl ventrolateral subregion of the ventromedial hypothalamus, IL infralimbic cortex, PAGl lateral subregion of the periaqueductal gray matter, PAGvl ventrolateral subregion of the periaqueductal gray matter. Scale bars: 100 μm. Data are shown as mean ± SEM, *$P < 0.05$. See also Supplementary Figs. 1, 2, and 3 and Supplementary Movies 1 and 2.

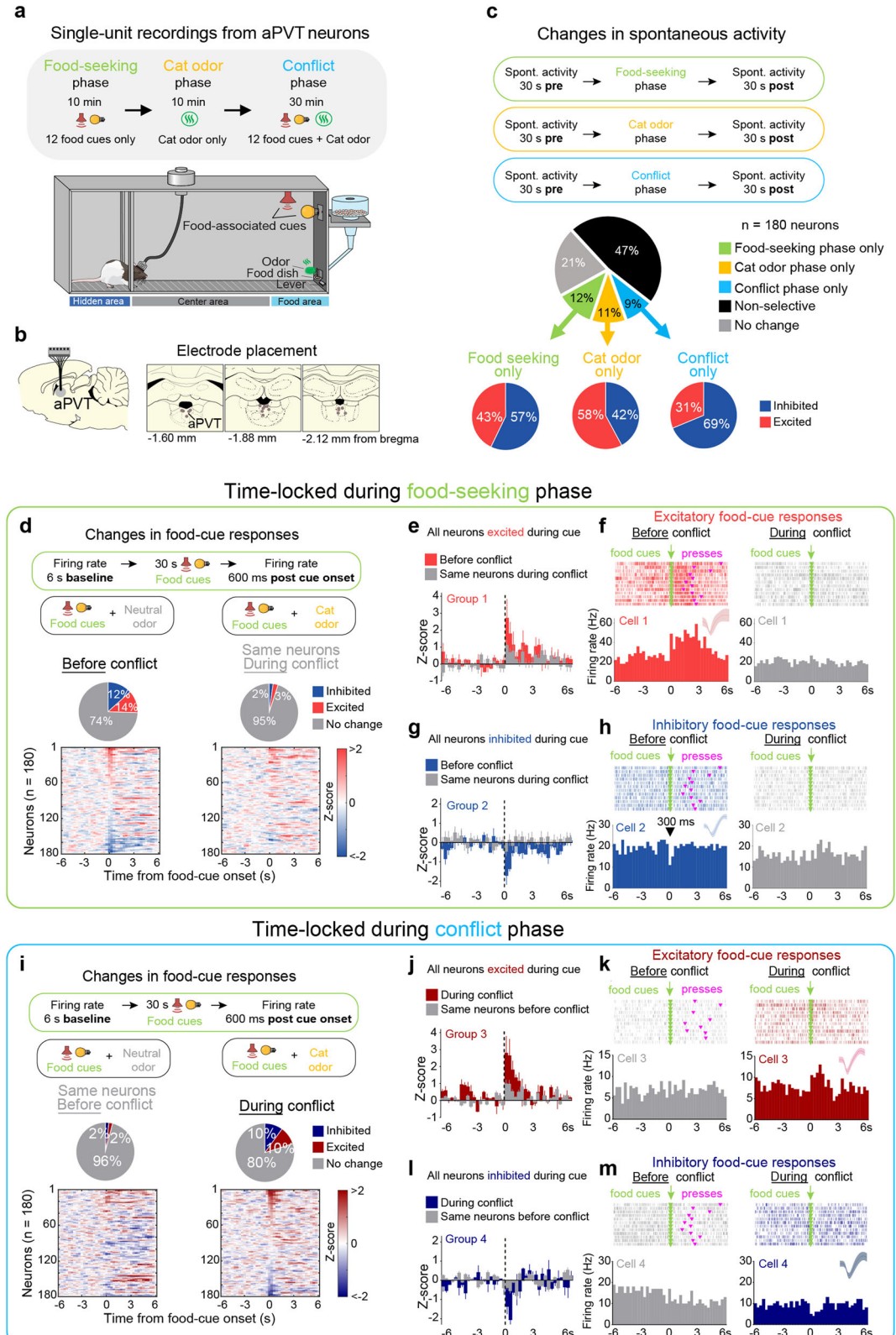

with a robust suppression in lever presses (Fig. 2e–h). When aPVT activity was time-locked to the onset of the food cues in the presence of cat odor, two distinct subsets of neurons emerged to exhibit either excitatory or inhibitory food-cue responses exclusively during the conflict phase (Fig. 2i–m). Food-cue responses were classified as fast (<1 s duration) or sustained responses (>1 s duration) and were similar in proportion during the food-seeking

and conflict phases (Supplementary Figure 5a–i). In addition to food cues, aPVT neurons also changed their firing rates in response to lever presses or rewarded dish entries, with most of the cells responding selectively to each one of these events (Supplementary Figure 4a–g). Together, these results demonstrate that cat odor and food-related information converge onto aPVT neurons, and suggest that the reduction in excitatory and

**Fig. 2 aPVT neurons change their firing rates during the approach-avoidance conflict test. a** Timeline and schematic of the approach-avoidance conflict test during single-unit recordings. **b** Diagram of the electrode placements in the aPVT. **c** (Top) Schematic of the spontaneous activity recordings. (Center) The percentage of aPVT neurons that changed their spontaneous activity exclusively in response to food-seeking phase, cat odor phase, conflict phase, in more than one phase (nonselective), or did not change. (Bottom) The percentage of aPVT-responsive neurons that were either excited or inhibited after each phase. **d** (Top) Schematic of the recordings during food-cue-evoked responses. (Center) The percentage of food-cue-responsive neurons selected before the conflict (left) are greater than that of the same neurons that are responsive during the conflict (right; Fisher's exact test; excitatory before the conflict: 25 neurons, excitatory during the conflict: 5 neurons, $P < 0.001$; inhibitory before the conflict: 22 neurons, inhibitory during the conflict: 4 neurons, $P < 0.001$). (Bottom) The normalized firing rate of individual aPVT neurons time-locked for food-cue onset before the conflict (left) and during the conflict (right, Z-score >2.58 for excitatory and <−1.96 for inhibitory responses, first two bins of 300 ms). **e**–**h** Average peristimulus time histograms (PSTHs) of all aPVT neurons showing **e** excitatory ($n = 21$ neurons) or **g** inhibitory ($n = 23$ neurons) food-cue responses before the conflict (red or blue) and the same neurons during the conflict (gray). Raster plot and PSTHs of representative aPVT neurons showing **f** excitatory or **h** inhibitory food-cue responses before the conflict (red or blue) and the same cells during the conflict (gray). **i** (Top) Same as d-Top but time-locked for food-cue onset during the conflict phase. The number of both excitatory and inhibitory food-cue-responsive neurons was greater during the conflict compared to before the conflict (Fisher's exact test, excitatory during the conflict: 18 neurons, excitatory before the conflict: 4 neurons, $P = 0.0026$; inhibitory during the conflict: 18 neurons, inhibitory before the conflict: 4 neurons, $P = 0.0026$). (Bottom) Same as d-Bottom but time-locked for food-cue onset during the conflict. **j**–**m** Average PSTHs of all aPVT neurons showing **j** excitatory ($n = 18$ neurons) or **l** inhibitory ($n = 18$ neurons) food-cue responses during the conflict (dark red or dark blue) and the same neurons before the conflict (gray). Raster plot and PSTHs of representative aPVT neurons showing **k** excitatory or **m** inhibitory food-cue responses during the conflict (dark red or dark blue) and the same cells before the conflict (gray). Inset: waveforms. $n = 180$ aPVT cells from 19 rats. See also Supplementary Figs. 4 and 5.

inhibitory food-cue responses observed during the conflict may result in an attenuation of food-seeking behavior.

**Pharmacological inactivation of aPVT neurons or chemogenetic inhibition of aPVT-NAc neurons biases behavior toward food seeking.** Our recordings described so far indicate that aPVT neurons either increase or decrease their firing rate during the conflict phase. To test whether aPVT activity is necessary to regulate an animal's behavior during the conflict, we pharmacologically inactivated aPVT neurons with the GABA$_A$ receptor agonist muscimol. Inactivation of aPVT reduced defensive behaviors and increased food approaching during the conflict test, without changing lever pressing (Fig. 3a–g). Importantly, inactivation of aPVT had no effect on defensive behaviors in rats exposed to cat odor alone (Supplementary Figure 6a). Furthermore, since previous work has shown that inactivation of aPVT does not affect cued food-seeking responses in a neutral context[12], this suggests an integrative role for aPVT during behavioral conflict. Neuroanatomical studies have shown that aPVT neurons send dense glutamatergic projections to the NAc[21], a key region in the regulation of reward seeking and motivation[22,23]. We, therefore, investigated if aPVT neurons that project to the NAc are involved in the modulation of behavioral responses during the conflict by specifically inactivating these cells during the conflict test. Rats were bilaterally infused into the NAc with a viral vector to express Cre recombinase in a retrograde manner, followed by an intra-aPVT infusion of a Cre-dependent viral vector to express either the inhibitory chemogenetic tool hM4Di or the control reporter mCherry. Clozapine-N-oxide (CNO)-induced inactivation of aPVT-NAc neurons reduced defensive responses and increased food-seeking behavior during the conflict (Fig. 3h–n and Supplementary Movie 3). Inactivation of this same pathway did not affect antipredator defensive behaviors in the absence of conflict (Supplementary Figure 6b) nor food-seeking responses in a neutral context[12]. Taken together, these data suggest that activity in aPVT-NAc projections acts to suppress food seeking during predator odor-induced conflict.

**aPVT neurons expressing corticotrophin-releasing factor are recruited during the conflict.** Corticotrophin-releasing factor (CRF) is the main physiological activator of the mammalian stress response and has been also implicated in food-seeking

regulation[27]. Two recent neuroanatomical studies in mice have identified a subpopulation of PVT neurons that co-expresses the neuropeptide CRF[28,29], but the physiological functions of these cells have not been explored. We, therefore, speculated that activity in CRF-expressing neurons in the aPVT (aPVT$^{CRF}$) could be the underlying mechanism that mediates suppression of food seeking during predator odor exposure. Using in situ hybridization, we first demonstrated that, as with mice, rats show a significant expression of CRF in PVT neurons, with similar levels of distribution (~35% of cells) along the anteroposterior (AP) aspect of PVT (Supplementary Figure 7a–c). Next, to specifically label aPVT$^{CRF}$ neurons, we used a viral vector with a gene promoter from a rat that favors the expression of Cre recombinase in CRF-positive neurons (AAV-CRF-Cre) in combination with a Cre-dependent eYFP reporter. This CRF-labeling approach has been successfully used in the previous studies[30], and was validated here for PVT neurons by using in situ hybridization (Supplementary Figure 7d). Exposure to cat odor led to a higher percentage of aPVT$^{CRF}$ neurons that were immunoreactive to cFos as well as a higher percentage of cFos-positive neurons that were labeled with the CRF reporter, when compared to neutral odor controls (Supplementary Figure 7e). To further investigate how aPVT$^{CRF}$ neurons respond during the conflict, we combined single-unit recordings with optogenetics to track the neuronal activity of photoidentified aPVT$^{CRF}$ neurons during the conflict. Rats expressing the light-activated cation channel channelrhodopsin (ChR2) selectively in aPVT$^{CRF}$ neurons were implanted with an optrode into the same region for optogenetic-mediated identification of aPVT$^{CRF}$ neurons at the end of the behavioral session (Fig. 4a, b). Among the aPVT-recorded neurons, 27% (26 out of 96) showed short-latency responses (<12 ms) to laser illumination and were classified as aPVT$^{CRF}$ neurons, whereas 63% (70 out of 96) showed either long-latency responses (>12 ms) or no responses to laser illumination and were classified as non-identified aPVT neurons (aPVT$^{non-ident}$) (Fig. 4c–f and see Methods). We examined the neuronal activity of aPVT neurons time-locked to the onset of the food cues to compare the response of the same cells before and during the conflict. Strikingly, the percentage of photoidentified aPVT$^{CRF}$ neurons that were responsive to food cues increased during the conflict, and this number was greater than that observed in the aPVT$^{non-ident}$ neurons (Fig. 4g). In addition, the magnitude of both excitatory and inhibitory food-cue responses in aPVT$^{CRF}$ neurons was higher during the conflict, when compared to the same cells before the conflict (Fig. 4h–i). Changes in spontaneous activity, lever-press responses, and dish-entry responses were similar between aPVT$^{CRF}$ and

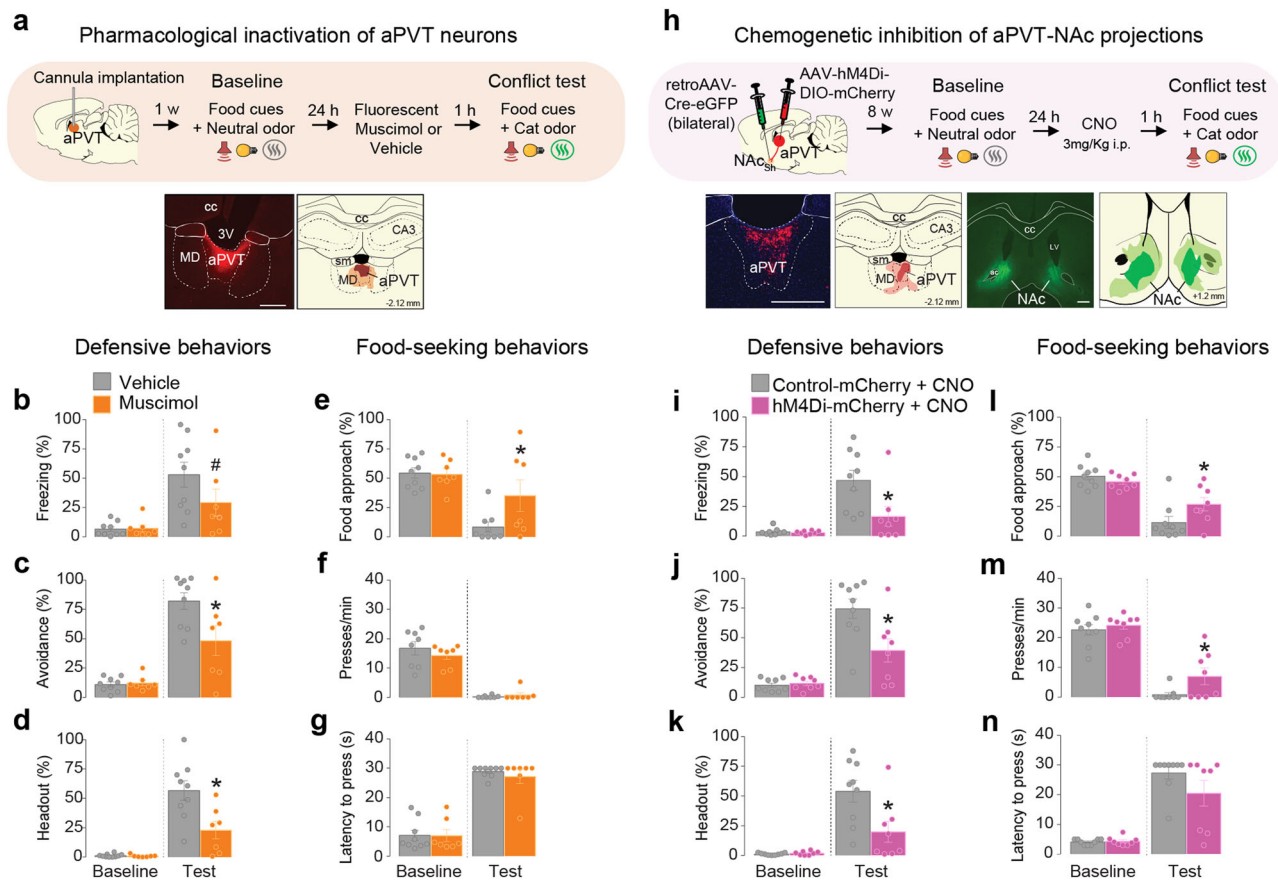

**Fig. 3 Pharmacological inactivation of aPVT neurons or chemogenetic inhibition of aPVT-NAc neurons biases behavior towards food seeking. a** (Top) Timeline of the approach-avoidance conflict test during pharmacological inactivation of aPVT neurons. (Bottom left) Representative micrograph showing the site of fluorescent muscimol microinjection into aPVT. (Bottom right) Orange areas represent the minimum (dark) and the maximum (light) spread of muscimol. **b–g** Muscimol inactivation of aPVT neurons (orange bars, $n = 7$) during the conflict test reduced the percentage of time rats spent exhibiting **c** avoidance ($F_{(1, 14)} = 5.59$, $P = 0.033$) and **d** head-out responses ($F_{(1, 14)} = 8.58$, $P = 0.011$), and increased **e** food-approach time ($F_{(1, 14)} = 3.588$, $P = 0.079$, with a Bonferroni planned comparison $P = 0.026$). Animals also exhibited a trend to reduce **b** freezing ($F_{(1, 14)} = 2.25$, $P = 0.155$, Bonferroni planned comparison $P = 0.092$). No changes were observed in **f** lever presses ($F_{(1, 14)} = 1.207$, $P = 0.29$) and **g** latency to press ($F_{(1, 14)} = 0.28$, $P = 0.60$), when compared to vehicle controls (gray bars, $n = 9$). **h** (Top) Timeline of the approach-avoidance conflict test during chemogenetic inhibition of aPVT-NAc neurons. (Bottom left) Representative micrograph showing the expression of hM4Di into the aPVT. Red areas represent the minimum (dark) and the maximum (light) viral expression into the aPVT. (Bottom right) Representative micrograph showing the site of microinjection of retrograde AAV-Cre-GFP into the NAc. Green areas represent the minimum (dark) and the maximum (light) viral expression into the NAc. **i–n** Chemogenetic inhibition of aPVT-NAc neurons (pink bars, $n = 8$) reduced the percentage of time rats spent exhibiting **i** freezing ($F_{(1, 15)} = 6.22$, $P = 0.024$), **j** avoidance ($F_{(1, 15)} = 10.58$, $P = 0.005$), and **k** head-out ($F_{(1, 15)} = 8.07$, $P = 0.012$) responses, and increased **l** food-approach time ($F_{(1, 15)} = 7.48$, $P = 0.015$) and **m** lever presses ($F_{(1, 15)} = 1.69$, $P = 0.21$, with Bonferroni planned comparison test $P = 0.040$), with no changes in the **n** latency to press ($F_{(1, 15)} = 2.50$, $P = 0.13$) during the conflict test, when compared to mCherry controls (gray bars, $n = 9$). cc corpus callosum, MD mediodorsal thalamus, 3V third ventricle, sm stria medullaris, CA3 hippocampal CA3 subregion, NAc nucleus accumbens, LV lateral ventricle, ac anterior commissure. Scale bars: 500 μm. Two-way repeated-measures ANOVA followed by Bonferroni post hoc test. Data are shown as mean ± SEM. *$P < 0.05$, #$P$ between 0.05 and 0.099. See also Supplementary Figs. 6 and 7 and Supplementary Movie 3.

aPVT[non-ident] neurons (Fig. 4j–l). Together, these results demonstrate that aPVT[CRF] neurons are preferentially recruited during conflicting situations in which both food- and predator-related cues occur together.

**aPVT[CRF] neurons regulate food-seeking and avoidance responses during the conflict.** To investigate whether aPVT[CRF] activity is necessary to regulate predator-threat vs. food-seeking conflict, we used chemogenetics to inactivate aPVT[CRF] neurons during the conflict (Fig. 5a). Rats were infused with a mix of two viral vectors into the aPVT to express either hM4Di or the control reporter mCherry exclusively in aPVT[CRF] neurons. Inactivation of aPVT[CRF] neurons reduced avoidance behavior and increased

food-seeking responses during the conflict test (Fig. 5b–g and Supplementary Movie 4). Consistent with our previous pharmacological experiments, inactivation of aPVT[CRF] soma had no effect on cat odor-induced defensive behavior or food seeking when these stimuli were presented in isolation nor locomotor activity (Supplementary Figure 6c, e–g). Because exposure to cat odor increased the activity of aPVT[CRF] neurons (Supplementary Figure 7e), we next explored whether optogenetic activation of these cells in a neutral context could mimic the food-seeking suppression induced by cat odor. Rats were infused with a mix of two viral vectors in the aPVT to express either ChR2 or the control reporter eYFP exclusively in aPVT[CRF] neurons. Photoactivation of aPVT[CRF] neurons in the absence of cat odor reduced lever presses and increased the latency to press after the

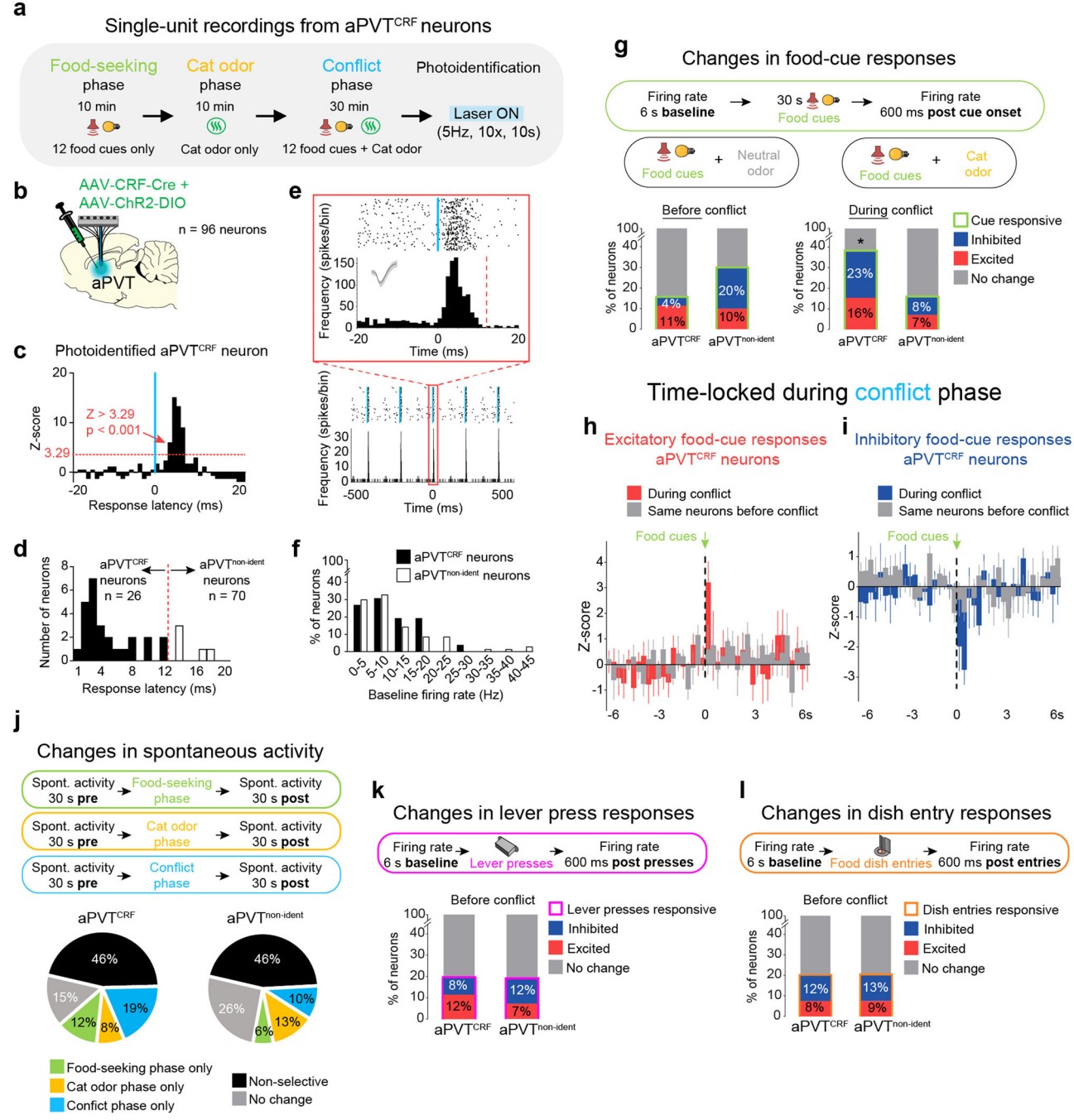

first food-cue presentation, when compared to eYFP controls (Fig. 5h–i and Supplementary Movie 5). To establish whether photoactivation of aPVT[CRF] neurons induces avoidance responses, we exposed the same animals to a real-time place preference task in which one side of the chamber was paired with laser illumination. Photoactivation of aPVT[CRF] neurons reduced both the percentage of time rats spent in the side of the chamber associated with the laser illumination and the total distance traveled on the same side, when compared to eYFP controls (Fig. 5j and Supplementary Movie 6). Photoactivation of aPVT[CRF] neurons had no effect on locomotor activity or freezing behavior (Fig. 5i, k). These results demonstrate that activation of aPVT[CRF] neurons can recapitulate both the behavioral suppression in food seeking and the increased avoidance responses observed during predator odor exposure.

**aPVT[CRF] neurons are interconnected with brain regions involved in the regulation of anti-predator and food-seeking behaviors.** To characterize the main anatomical outputs of aPVT[CRF] neurons, we infused a mix of two viral vectors to express eGFP specifically in aPVT[CRF] neurons, and labeled their axonal fibers with immunohistochemistry (Fig. 6a, b). We found that aPVT[CRF] neurons send projections to the dorsolateral subregion of the bed nucleus of stria terminalis, central nucleus of the amygdala, dorsomedial hypothalamus (DMH) and VMH, suprachiasmatic nucleus of hypothalamus, thalamic reticular nucleus (Rt), and ventral subiculum, with the most dense projections being observed along the AP extension of the nucleus accumbens shell (NAcSh) (Fig. 6c–j). Next, to identify the main anatomical inputs to aPVT[CRF] neurons, we used a virus-mediated retrograde trans-synaptic tracing method[31]. Rats were infused with a mix of three

**Fig. 4 aPVT^CRF neurons are recruited during the approach-avoidance conflict test. a** Timeline of the approach-avoidance conflict test during single-unit recordings from photoidentified aPVT^CRF neurons. **b** Diagram showing the injection of viral mix containing AAV-CRF-Cre and AAV-ChR2-DIO, and the implantation of optrode in aPVT. **c–e** Photoidentification of aPVT^CRF neurons. **c** Representative aPVT^CRF neuron responsive to laser illumination (Z-score >3.29, P < 0.001, red dotted line, see details in Methods). **d** Cells with photoresponse latencies <12 ms were classified as aPVT^CRF neurons (black bars, n = 26 out of 96 recorded neurons), whereas cells with photoresponse latencies >12 ms (white bars, n = 5 neurons) or non-responsive to the laser (n = 65 neurons, not shown) were classified as non-identified aPVT neurons (aPVT^non-ident, n = 70 out of 96 recorded neurons). **e** Raster plot and firing rate of a representative aPVT^CRF neuron responding to a 5 Hz train of laser stimulation. Inset: Raster plot and firing rate time-locked for laser onset. Vertical blue bars: laser onset. Bins of 1 ms. **f** Relative frequency histogram showing the baseline firing rate of aPVT^CRF neurons and aPVT^non-identif neurons. **g** (Top) Schematic of the food-cue-evoked responses. (Bottom) Percentage of aPVT^CRF and aPVT^non-ident neurons that were responsive to food cues (green bars) before (left) and during the conflict (right). aPVT^CRF neurons showed more food-cue responses during the conflict test, when compared to aPVT^non-ident neurons (Fisher's exact test; aPVT^CRF neurons: 39%, 10 out of 26; aPVT^non-ident neurons: 15%, 11 out of 70 neurons, P = 0.039). **h–i** Average peristimulus time histograms of all photoidentified aPVT^CRF neurons showing **h** excitatory or **i** inhibitory food-cue responses during the conflict (red or blue bars, respectively) or the same neurons before the conflict (gray bars). **j** (Top) Schematic of the spontaneous activity recordings. (Bottom) Percentage of aPVT^CRF and aPVT^non-ident neurons that changed their baseline spontaneous activity (30 s pre vs. 30 s post) exclusively in food-seeking phase, cat odor phase, conflict phase (30 min), in more than one phase (nonselective), or did not change. No differences were observed between the two groups (Fisher's exact test, all P's > 0.05). **k** (Top) Schematic of the recordings during lever presses-evoked responses. (Bottom) Percentage of aPVT^CRF and aPVT^non-ident neurons that were responsive to lever presses (pink bars) before the conflict phase. **l** (Top) Schematic of the recordings during dish-entry-evoked responses. (Bottom) Percentage of aPVT^CRF and aPVT^non-ident neurons that were responsive to rewarded dish entries (orange bars) before the conflict phase. No differences were observed between the two groups (Fisher's exact test, all P's > 0.05). A total of eight rats were used.

viral vectors into the aPVT to express TVA and rabies glycoprotein exclusively in aPVT^CRF neurons. Three weeks later, a glycoprotein-deleted pseudotyped rabies virus was infused into the same aPVT region to enable the retrograde monosynaptic labeling of efferents to aPVT^CRF neurons (Fig. 6k, l and see Methods). We found that aPVT^CRF neurons receive direct inputs from PL, Rt, median preoptic area, VMH, lateral hypothalamus (LH), and PAGdm-dl (Fig. 6m–r). These results show that aPVT^CRF neurons are connected with areas of the brain that have been implicated in the regulation of food seeking (e.g., NAc, PL, LH)[14] and anti-predator defensive responses (e.g., VMHdm, PAGdm-dl)[26].

**Activation of aPVT^CRF–NAc projections reduces food seeking, increases avoidance, and evokes target-dependent synaptic responses in the NAc in vitro.** Our neuroanatomical tracing findings demonstrated that aPVT^CRF neurons project densely to the NAc. To further establish whether increased activity in the aPVT^CRF-NAc pathway suppresses food seeking, we infused a mix of two viral vectors into the aPVT to express either ChR2 or the control reporter eYFP exclusively in aPVT^CRF neurons, and implanted bilateral optical fibers in the NAc to illuminate aPVT^CRF fibers in this region. Similar to photoactivation of aPVT^CRF somata, photoactivation of aPVT^CRF projections in the NAc reduced lever presses and increased the latency to press after the first food-cue presentation, when compared to eYFP controls (Fig. 7a, b and Supplementary Movie 7). Photoactivation of aPVT^CRF projections in the NAc also produced avoidance responses in the real-time place preference task, without changing locomotor activity or inducing freezing behavior (Fig. 7b–d and Supplementary Movie 8). These findings suggest that increased activity in the aPVT-NAc pathway is sufficient to both suppress food-seeking and induce avoidance responses.

To further probe the functional connectivity of aPVT^CRF afferents onto distinct types of NAc neurons, we performed whole-cell recordings from putative medium spiny neurons (MSNs) and cholinergic interneurons (CINs) in rat brain slices expressing ChR2-eYFP in aPVT^CRF afferents (Fig. 8a–c). Optogenetic stimulation led to fast-latency evoked excitatory post-synaptic currents (EPSCs) that were completely blocked by the α-amino-3-hydroxy-5-methyl-4-isoxazolepropionic acid (AMPA) receptor antagonist 2,3-dihydroxy-6-nitro-7-sulfamoyl-benzoqui-noxaline-2,3-dione (NBQX) (Fig. 8d, e). Performing sequential recordings from mixed pairs of neurons, we found that eight out of nine pairs of EPSCs recorded from MSNs were larger than

those recorded from CINs (Fig. 8f, g). Furthermore, aPVT^CRF synaptic responses onto CINs displayed short-term facilitation, while responses onto MSNs displayed modest short-term depression at steady-state (Fig. 8h, i). While on-terminal optical stimulation may affect short-term plasticity recorded under our experimental conditions[32], our data suggest that the target-dependent synaptic properties of aPVT^CRF-NAc afferents lead to distinct temporal patterns of activation in two major subclasses of NAc neurons.

**Inhibition of VMH projections to aPVT reduces defensive behaviors during the conflict.** Our cFos experiments above demonstrated that cat odor exposure activates both VMHdm (Fig. 1l) and aPVT^CRF neurons (Supplementary Figure 6f). In addition, our tracing study showed that aPVT^CRF neurons receive direct monosynaptic inputs from VMH (Fig. 6p). We then hypothesized that VMH neurons convey predator-related information to aPVT to regulate food-approach vs. predator-threat avoidance responses. To investigate whether the VMH-aPVT pathway is activated by predator threat, we infused a retrograde virus into the aPVT to label VMH neurons that project to aPVT and performed cFos immunofluorescent staining in VMH sections from rats exposed to either cat odor or neutral odor (Fig. 9a). Exposure to cat odor increased both the percentage of VMH-aPVT neurons that were immunoreactive to cFos as well as the percentage of cFos-positive neurons in VMH that project to aPVT, when compared to neutral odor-exposed controls (Fig. 9b, c). To investigate if VMH neurons form functional synapses with aPVT neurons that project to the NAc, we performed whole-cell recording in aPVT slices from rats previously injected with both the retrograde tracer cholera toxin b (CTB) into the NAc and ChR2 into the VMH. We observed that photoactivation of VMH synaptic afferents in the aPVT elicited large EPSCs in aPVT-NAc neurons, which were fully blocked by NBQX (Fig. 9d–f). These results demonstrate that VMH can indirectly modulate the activity of NAc neurons through its glutamatergic projections to the aPVT.

Next, to explore the role of the VMH-aPVT pathway during the conflict, we used an intersectional viral vector approach to express hM4Di or the control reporter mCherry in VMH-aPVT neurons and chemogenetically inactivated these neurons during the conflict test (Fig. 9g). CNO-induced inactivation of VMH-aPVT neurons attenuated defensive responses (Fig. 9h–j and Supplementary Movie 9), but did not change the food-seeking suppression induced by cat odor, when compared to mCherry controls (Fig. 9k–m).

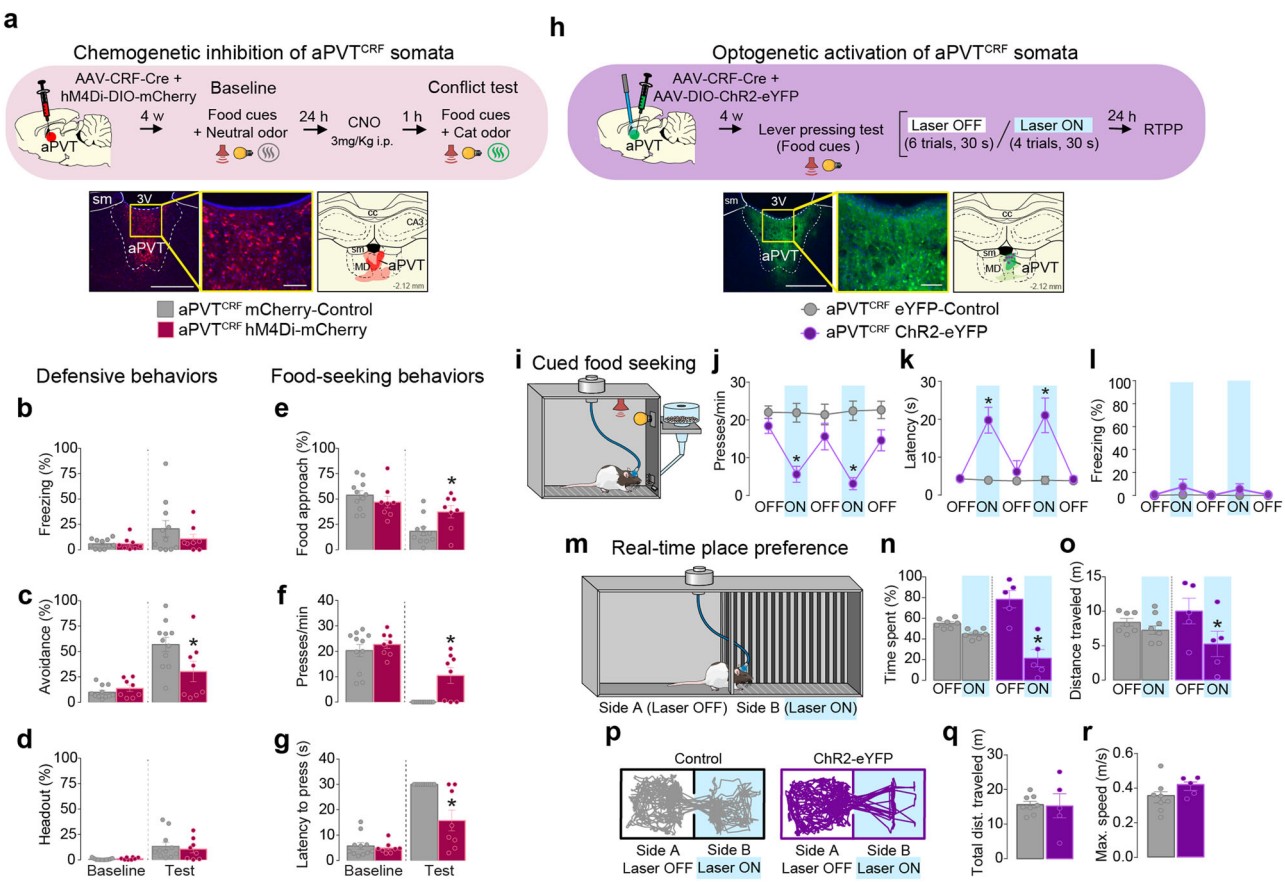

**Fig. 5 Activity in aPVT^CRF neurons bidirectionally regulates food seeking and avoidance behavior. a** (Top) Timeline of the conflict test during chemogenetic inhibition of aPVT^CRF neurons. (Bottom left) Representative micrograph showing the viral expression in aPVT. (Bottom center) High magnification of the same micrograph. (Bottom right) Red areas represent the minimum (dark) and the maximum (light) viral expression. **b–g** Chemogenetic inhibition of aPVT^CRF neurons (red wine bars, $n = 8$) reduced the percentage of time rats spent exhibiting **c** avoidance ($F_{(1, 17)} = 6.41$, $P = 0.021$), and increased **e** food-approach time ($F_{(1, 17)} = 13.9$, $P = 0.0017$) and **f** lever presses ($F_{(1, 17)} = 4.62$, $P = 0.046$), with a reduction in the **g** latency to press ($F_{(1, 17)} = 11.35$, $P = 0.0036$) during the conflict test, when compared to mCherry controls (gray bars, $n = 11$). No changes were observed in **b** freezing ($F_{(1, 17)} = 0.8813$, $P = 0.36$) and **d** head-out ($F_{(1, 17)} = 0.34$, $P = 0.56$). **h** (Top) Timeline of the cued food-seeking test during optogenetic activation of aPVT^CRF neurons. (Bottom left) Representative micrograph showing the viral expression in aPVT. (Bottom center) High magnification of the same micrograph. (Bottom right) Green areas represent the minimum (dark) and the maximum (light) viral expression. Purple dots: optical fiber tips. **i** Schematic of the cued food-seeking test. **j–l** Optogenetic activation of aPVT^CRF neurons (purple solid circles, $n = 6$) reduced the **j** number of lever presses ($F_{(4, 48)} = 11.4$, $P < 0.001$) and increased **k** the latency to press the lever ($F_{(4, 48)} = 14.21$, $P < 0.001$), when compared to eYFP controls (gray circles, $n = 8$). No difference was found in **l** the percentage of time rats spent freezing during the illumination ($F_{(4, 48)} = 1.691$, $P = 0.168$). Blue shaded area represents laser-on trials (20 Hz, 5 ms pulse width, 10 mW, 30 s duration). Each circle represents the average of two consecutive trials. **m** Schematic of the real-time place preference test. **n–r** Photoactivation of aPVT^CRF neurons (purple bars, $n = 5$) reduced both **n** the percentage of time spent ($F_{(1, 11)} = 7.79$, $P = 0.018$) and **o** the distance traveled ($F_{(1, 11)} = 7.238$, $P = 0.021$) in the side of the chamber paired with laser stimulation (Side B), when compared to eYFP controls (gray bars, $n = 7$). **p** Representative tracks. No difference was found in locomotor activity measured as **q** total distance traveled ($P = 0.911$, $t = 0.11$) and **r** maximum speed ($P = 0.183$, $t = 1.42$) during the session. Blue shaded areas represent the sum of all laser-on epochs (20 Hz, 5 ms pulse width, 10 mW). sm stria medullaris, 3V third ventricle, cc corpus callosum, MD mediodorsal thalamus, CA3 hippocampal CA3 subregion. Scale bars: 500 μm; inset scale bars: 100 μm. **b–g, j–l, n, o** Two-way repeated-measures ANOVA followed by Bonferroni test. **q, r** Unpaired Student's $t$ test. Data are shown as mean ± SEM. *$P < 0.05$. See also Supplementary Movies 4, 5, and 6.

Chemogenetic inactivation of these same neurons had no effect on anti-predator defensive behaviors in the absence of conflict (Supplementary Figure 6d). Finally, to test if photoactivation of VMH-aPVT projections is sufficient to reduce lever presses and induce avoidance responses similar to the cat odor-induced conflict, we infused a viral vector to express either ChR2 or the control reporter eYFP into the VMH and implanted an optical fiber into the aPVT. We found that photoactivation of VMH-aPVT projections reduced cued food seeking in a neutral context and induced avoidance responses during the real-time place preference task, without promoting freezing behavior (Supplementary Figure 8a–c). Using a modified version of the cued food-seeking test (shuttle food-seeking test) in which rats had to alternate between two sides of the chamber to press for food in the side signaled by the food cue, we demonstrated that photoactivation of VMH-aPVT projection at the onset of the food area entry was sufficient to suppress lever presses (Supplementary Figure 8d and Supplementary Movie 10). Overall, these results confirm our hypothesis that VMH neurons transmit predator-threat information to aPVT to modulate approach vs. avoidance responses (Fig. 10).

## Discussion

Laboratory rats exhibit strong defensive responses toward cat odor, despite the fact that they have never encountered a cat[26,33].

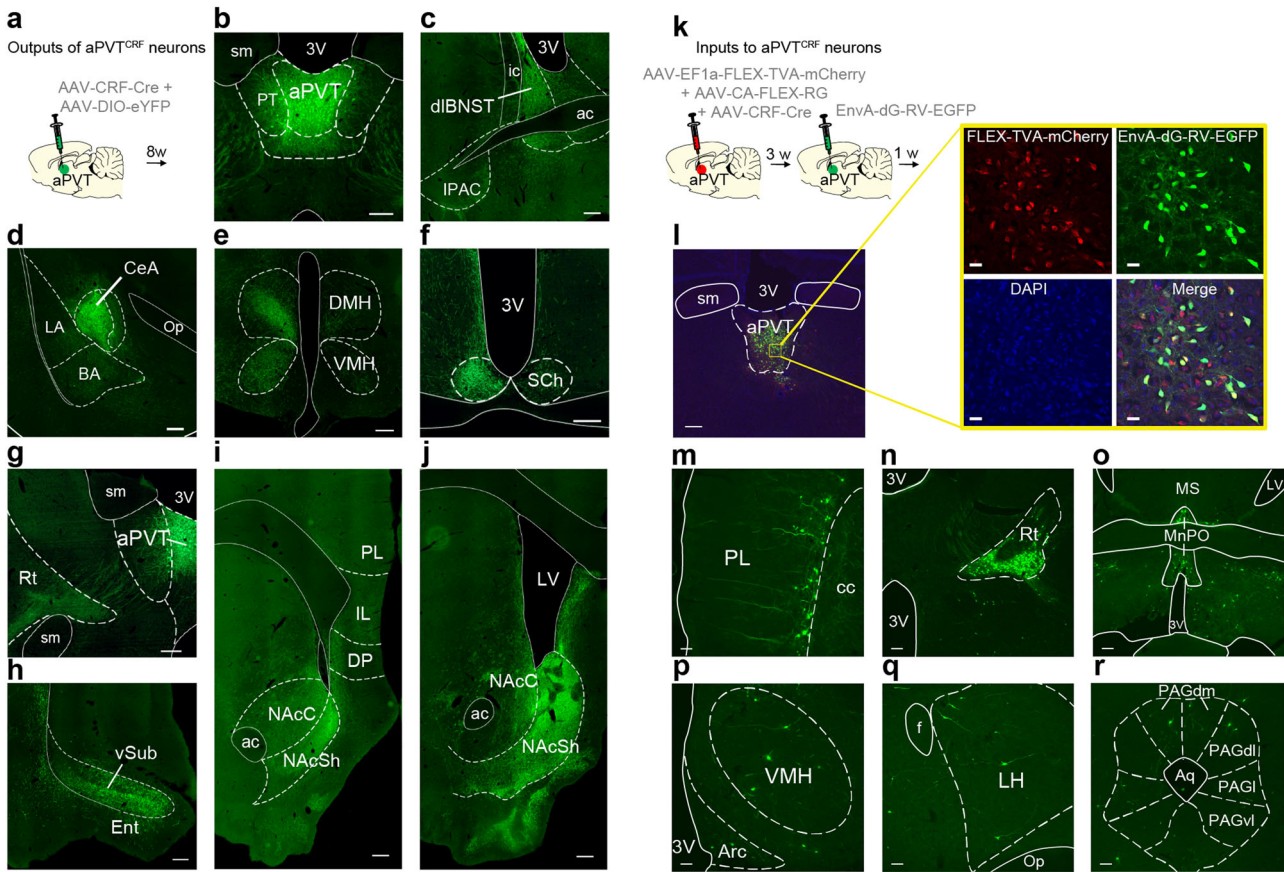

**Fig. 6 Afferent and efferent connections of aPVT^CRF neurons across the brain. a** Schematic of intra-aPVT microinjection of a viral mix containing AAV-CRF-Cre and AAV-DIO-eYFP for specific labeling of aPVT^CRF neurons and their fibers in output regions. **b–j** Representative micrographs showing the expression of eYFP in **b** aPVT^CRF neurons and their GFP-labeled fibers located in the **c** dorsolateral subregion of the bed nucleus of stria terminalis (dlBNST), **d** central nucleus of the amygdala (CeA), **e** dorsomedial hypothalamus (DMH), and ventromedial hypothalamus (VMH), **f** suprachiasmatic nucleus of hypothalamus (SCh), **g** reticular thalamic nucleus (Rt), **h** ventral subiculum (vSub), and both **i** the anterior and **j** the posterior subregions of the nucleus accumbens shell (NAcSh). Scale bars: 500 μm. This experiment was repeated in three rats with similar results. **k** Diagram of the viral vector infusions used for monosynaptic retrograde tracing of aPVT^CRF neurons. **l** Representative micrograph of the injection site in aPVT showing the expression of helper virus (red) and rabies virus (green) under the control of AAV-CRF-Cre virus. Scale bar: 200 μm. Inset: High magnification of the same micrograph. Scale bars: 25 μm. **m–r** Monosynaptic efferents to aPVT^CRF neurons identified by the rabies virus, including **m** prelimbic cortex (PL), **n** reticular thalamic nucleus (Rt), **o** median preoptic area (MnPO), **p** ventromedial hypothalamus (VMH), **q** lateral hypothalamus (LH), and the **r** dorsomedial and dorsolateral subregions of the periaqueductal gray matter (PAGdm/PAGdl). Scale bars: 100 μm. This experiment was repeated in two rats with similar results. 3V third ventricle, sm stria medullaris, PT paratenial nucleus of thalamus, ic internal capsule, ac anterior commissure, IPAC interstitial nucleus of the posterior limb of the anterior commissure, LA lateral amygdala, BA basal amygdala, op. optic tract, Ent entorhinal cortex, PL prelimbic cortex, IL infralimbic cortex, DP dorsal peduncular cortex, NAcC nucleus accumbens core, cc corpus callosum, LV lateral ventricle, MS medial septum, Arc arcuate nucleus of the hypothalamus, Op optical tract, f fornix, MRe mammillary recess of the third ventricle, cp cerebral peduncle, Aq cerebral aqueduct, PAGl lateral subregion of the periaqueductal gray matter, PAGvl ventrolateral subregion of the periaqueductal gray matter.

Exposure to cat odor also induces a reduction in food-seeking responses[3], but few studies have attempted to determine which neural circuits regulate the behavioral conflict between seeking food and avoiding a predator threat. Here, we designed an ethologically relevant experimental model to investigate this question. In our approach-food vs. avoid predator-threat conflict test, rats exhibited strong defensive responses and suppression of food-seeking behavior during the concomitant presentation of cat odor and food cues. We identified a subpopulation of aPVT neurons that express CRF and are preferentially recruited during the conflict. Neuroanatomical analyses revealed that aPVT^CRF neurons are interconnected with brain regions associated with foraging and anti-predator defenses. Using both loss- and gain-of-function manipulations, we found that activity in the aPVT^CRF-NAc pathway attenuates food seeking, whereas activity in the VMH-aPVT pathway mediates defensive responses exclusively during the conflict. Our results revealed an aPVT^CRF

circuit that integrates food signals with predator information and serves to regulate approach vs. avoidance responses during the conflict.

Compared with previous conflict models using shock-based fear conditioning or predator-like robot paradigms[34–38], our conflict test has some advantages: (i) cat saliva elicits a large range of defensive responses that, in contrast to shock-induced freezing responses in a small chamber, better resemble the wide repertoire of defensive behaviors expressed in natural environments; (ii) rats' defensive responses to cat saliva do not habituate across the session, as has been observed with artificial predator threats; (iii) cat saliva activates the same neural circuits mobilized by natural predator threats, such as the medial hypothalamic defensive system[24,39]. In addition, different from synthetic predator odor cues (e.g., 2,4,5-trimethylthiazoline), rats show increased defensive responses when re-exposed to the context in which they have previously experienced the cat saliva (contextualization)[3], making

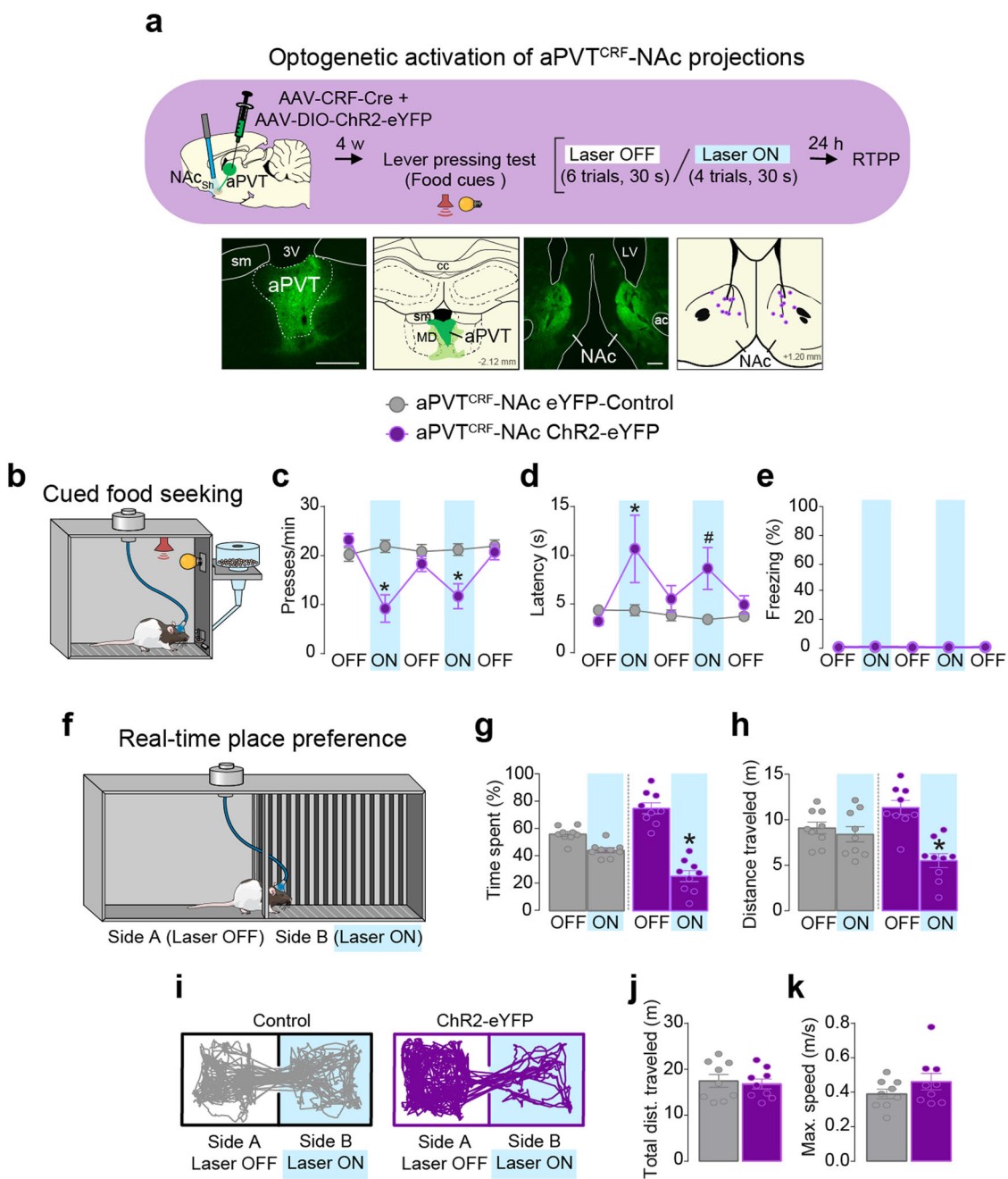

**Fig. 7 Photoactivation of aPVT^CRF-NAc projections suppresses food-seeking and induces avoidance behavior. a** (Top) Timeline of the cued food-seeking test during optogenetic activation of aPVT^CRF-NAc projections. (Bottom left) Representative micrograph showing the expression of Cre-dependent channelrhodopsin (AAV-DIO-ChR2-eYFP) under the control of AAV-CRF-Cre in aPVT. (Bottom left-center) Green areas represent the minimum (dark) and the maximum (light) viral expression in the aPVT. (Bottom right-center) Representative micrograph showing aPVT^CRF fibers in the NAc. (Bottom right) Purple dots represent the location of the optical fiber tips in the NAc. sm stria medullaris, 3V third ventricle, cc corpus callosum, MD mediodorsal thalamus, ac anterior commissure. **b** Schematic of the cued food-seeking test. **c–e** Optogenetic activation of aPVT^CRF-NAc projections (purple circles, $n = 10$) reduced **c** the number of lever presses ($F_{(4, 72)} = 11.49$, $P < 0.001$) and increased **d** the latency to press the lever ($F_{(4, 72)} = 2.379$, $P = 0.059$ with Bonferroni planned comparison test $P = 0.012$ for the first laser-on block and $P = 0.054$ for the second laser-on block), when compared to eYFP controls (gray circles, $n = 10$). No difference was found in **e** the percentage of time rats spent freezing during the illumination ($F_{(1, 16)} = 4.316$, $P = 0.54$). Blue shaded area represents laser-on trials (20 Hz, 5 ms pulse width, 15 mW, 30 s duration). Each circle represents the average of two consecutive trials. **f** Schematic of the real-time place preference test. **g–k** Photoactivation of aPVT^CRF-NAc projections (purple bars, $n = 9$) reduced both **g** the percentage of time spent ($F_{(1, 16)} = 18.73$, $P < 0.001$) and **h** the distance traveled ($F_{(1, 16)} = 16.78$, $P < 0.001$) in the side of the chamber paired with laser stimulation (Side B), when compared to eYFP controls (gray bars, $n = 9$). **i** Representative tracks. No difference was found in locomotor activity measured as **j** total distance traveled ($P = 0.711$, $t = 0.378$) and (**k**) maximum speed ($P = 0.212$, $t = 1.301$) during the session. Blue shaded areas represent the sum of all laser-on epochs (20 Hz, 5 ms pulse width, 15 mW). **c–e, g, h** Two-way repeated-measure ANOVA followed by Bonferroni test. **j, k** Unpaired Student's $t$ test. Data are shown as mean ± SEM. *$P < 0.05$, #$P$ between 0.05 and 0.099. See also Supplementary Movies 7 and 8.

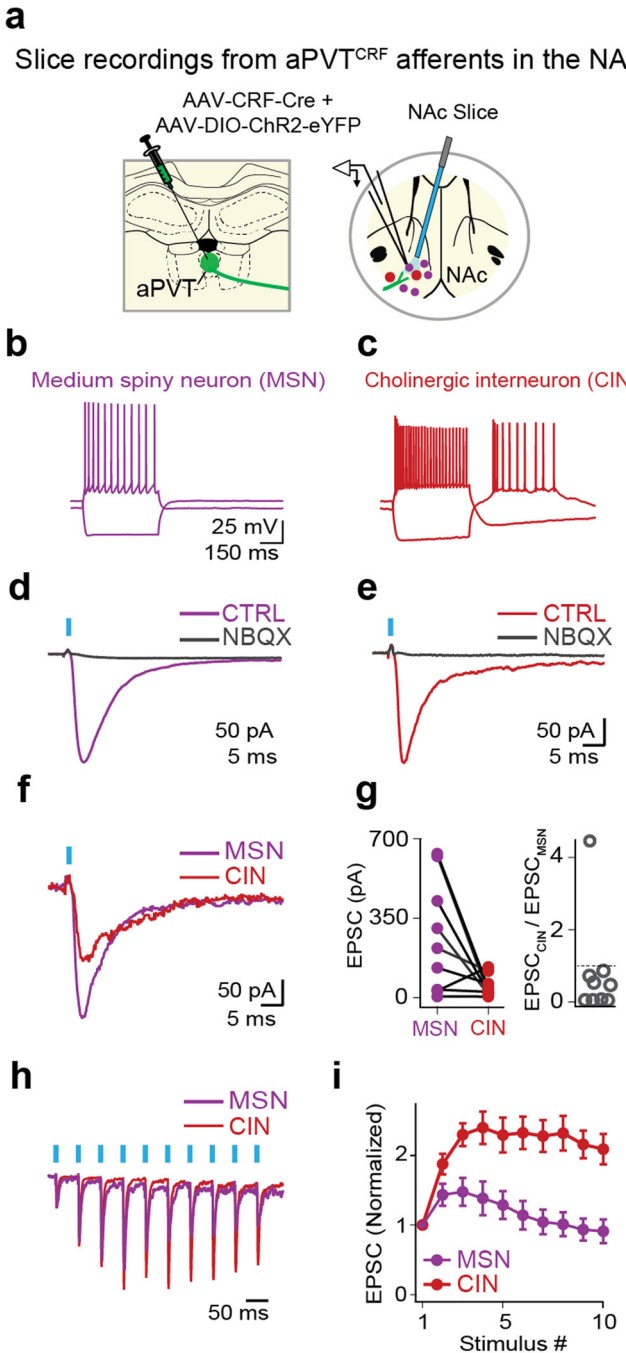

**a**

Slice recordings from aPVT$^{CRF}$ afferents in the NAc

AAV-CRF-Cre +
AAV-DIO-ChR2-eYFP     NAc Slice

aPVT          NAc

**b**  Medium spiny neuron (MSN)    **c**  Cholinergic interneuron (CIN)

25 mV
150 ms

**d**          CTRL          **e**          CTRL
NBQX                       NBQX

50 pA          50 pA
5 ms           5 ms

**f**          MSN
CIN

50 pA
5 ms

**g**

**h**          MSN
CIN

50 ms

**i**

Fig. 8 aPVT$^{CRF}$ neurons mediate target-dependent synaptic transmission in the NAc in vitro. **a** Schematics showing viral injection site in aPVT and recording site in NAc in rat brain slices expressing ChR2-eYFP in aPVT$^{CRF}$ afferents. **b**, **c** Representative whole-cell recordings of a putative **b** medium spiny neuron (MSN, purple) and **c** cholinergic interneuron (CIN, red) showing responses to hyperpolarizing and depolarizing current steps ($-300$ and $200$ pA, $500$ ms). CINs were distinguished from MSNs by larger diameter cell bodies and rebound firing following a hyperpolarizing current step. **d**, **e** Representative examples of optically evoked excitatory postsynaptic currents in an **d** MSN and a **e** CIN prior to and following the application of the AMPA receptor antagonist NBQX. **f** Representative examples of optically evoked excitatory postsynaptic currents (EPSCs) in an MSN (purple) and a neighboring CIN (red) recorded sequentially (LED duration: 1 ms). **g** (Left) Summary data quantifying optically evoked EPSC amplitudes in sequentially recorded pairs of MSNs and CINs ($n = 9$ pairs, three rats), indicating larger EPSCs in MSNs (paired Student's $t$ test, $P = 0.038$, $t = 2.483$). (Right) For the same pairs, graph plots $EPSC_{CIN}$/$EPSC_{MSN}$ ratios. **h** Representative synaptic responses in sequentially recorded MSN (purple) and a CIN (red) evoked by a 20 Hz stimulus train. Responses are normalized to the amplitude of $EPSC_1$. **i** Summary data quantifying EPSC amplitudes of MSNs (purple) and CINs (red) normalized to $EPSC_1$ ($n = 9$ pairs). Short-term synaptic plasticity (quantified as $EPSC_{8-10}$/$EPSC_1$) showed significant target-dependent differences (MSN: $0.95 \pm 0.17$; CIN: $2.19 \pm 0.22$, paired Student's $t$ test, $P = 0.0014$, $t = 4.79$). Scale bars: $500$ µm. Data are shown as mean ± SEM. Blue bars indicate the timing of LED stimuli.

integrates anti-predator defenses and food-related information to regulate the competing drives of approaching food and avoiding potential threats. Our results showing that aPVT neurons change their firing rate in response to both food cues and predator odor indicate that this thalamic region is a potential candidate for this integrative role. Consistently, all aPVT inactivation we performed in this study biased rat's behavior toward food seeking during the conflict test. These results agree with previous studies using shock-based fear conditioning and counterconditioning paradigms to demonstrate that PVT activity is critical to regulate the competition between appetitive and aversive behaviors[34,35]. The lack of effect of aPVT inactivation on innate defensive responses in the absence of conflict seems at odds with previous studies showing that inactivation of PVT impairs the retrieval of conditioned fear responses[8,9]. However, this apparent discrepancy may be attributed to differences in the neural circuits that mediate anti-predator and conditioned defensive responses[42], or alternatively, differences in the target of the inactivation between the current study (aPVT) and the previous reports (pPVT), as these two subregions of PVT exhibit genetically distinct subtypes of cells and neuroanatomical connections[17,21,43].

Similar to recent reports in mice[28,29], here we characterized in rats a subpopulation of PVT neurons that express CRF. Prior studies have demonstrated that exposure to predator odor activates CRF transmission[44], and either endogenous release or systemic administration of CRF reduces food intake and induces defensive behaviors in different species[27]. Our observation that aPVT$^{CRF}$ neurons are activated by cat odor, but indispensable for food-seeking suppression exclusively during the conflict, argues in favor of a critical role of these cells in regulating behaviors with opposite motivational drives. In support of this idea, we found that aPVT$^{CRF}$ neurons send dense glutamatergic projections to the NAcSh, a region implicated in reward-seeking motivation[22,23], and photoactivation of the aPVT$^{CRF}$-NAc pathway in a neutral environment suppresses food-seeking and elicits avoidance responses in a way that resembles the effects of cat odor exposure. This reduction in food seeking by activation of aPVT$^{CRF}$-NAc projections is consistent with previous studies showing that

our model suitable to investigate how unsafe contexts affect foraging behavior. Furthermore, compared to other sources of cat odor (e.g., cat-worn collar, cat fur/skin, impregnated cloth), cat saliva has the advantage of being quantifiable and relatively easy to collect and store (see Methods).

Our observation that rats display a series of defensive behaviors and strong suppression of food-seeking responses during the conflict test is consistent with previous studies in natural environments[3]. Under the risk of predation, individuals delay food-seeking responses and allocate more time and effort to avoid potential harm[3,40]. Given the often fatal outcome of failing to avoid a predator, animals must adjust their foraging behavior primarily to the predation risk and only secondarily to starvation[41]. Therefore, in risky environments, animals are expected to fine-tune their gradient of defensive responses to enable foraging behavior[2]. These considerations suggest the existence of a common node that

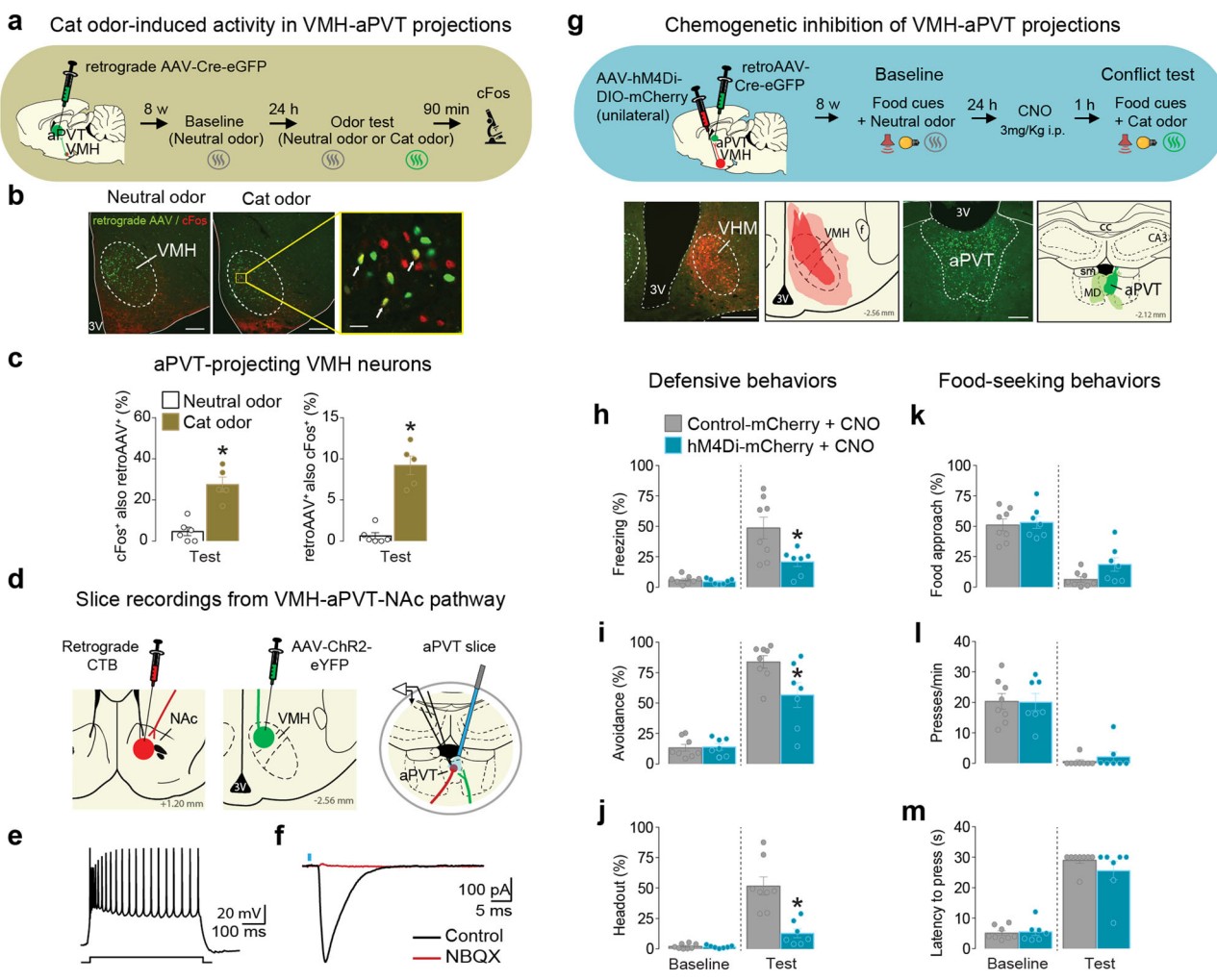

**Fig. 9 Chemogenetic inhibition of VMH-aPVT neurons attenuates defensive behaviors during the conflict test. a–c** Cat odor exposure activated VMH-aPVT neurons. **a** Timeline of the cat odor-induced neuronal activity test. **b** Representative micrographs showing VMH-aPVT neurons (green) expressing immunoreactivity to cFos (red) in both neutral odor (left) or cat odor (right) groups. Inset: white arrows showing examples of co-labeled cells. **c** Cat odor exposure (brown bars, $n = 5$) significantly increased (left) the percentage of VMH-aPVT cells that were cFos-positive and (right) the percentage of cFos-positive cells that were aPVT projecting, when compared to neutral odor controls (white bars, $n = 6$; unpaired Student's $t$ test; left, $P < 0.001$, $t = 7.64$; right, $P < 0.001$, $t = 5.78$). **d–f** Photoactivation of VMH afferents induces monosynaptic EPSCs in NAc-projecting aPVT neurons in vitro. **d** Schematics showing CTB infusion in NAc, viral vector AAV-ChR2-eYFP infusion in VMH, and slice recordings from NAc-projecting aPVT neurons. **e** Action potentials evoked by a depolarizing current step in a NAc-projecting aPVT neuron. **f** For the same cell, optically evoked EPSC (black) was completely blocked following bath application of the AMPA receptor antagonist NBQX (red). Similar findings were made in nine additional neurons (two rats). **g** (Top) Timeline of the conflict test during chemogenetic inhibition of VMH-aPVT neurons. (Bottom left) Representative micrograph showing the unilateral expression of hM4Di in VMH. (Bottom left-center) Red areas represent the minimum (dark) and the maximum (light) viral expression. (Bottom right-center) Representative micrograph showing the expression of retrograde AAV-Cre-GFP in aPVT. (Bottom right) Green areas represent the minimum (dark) and the maximum (light) viral expression. **h–m** Chemogenetic inhibition of VMH-aPVT neurons (blue bars, $n = 7$) reduced the percentage of time rats spent exhibiting **h** freezing ($F_{(1, 13)} = 7.35$, $P = 0.017$), **i** avoidance ($F_{(1, 13)} = 4.59$, $P = 0.051$ with Bonferroni planned comparison test $P = 0.005$) and **j** head-out responses ($F_{(1, 13)} = 18.64$, $P < 0.001$). No changes were observed in **k** food-approach time ($F_{(1, 13)} = 0.92$, $P = 0.35$), **l** lever press ($F_{(1, 13)} = 0.18$, $P = 0.67$) and **m** latency to press ($F_{(1, 13)} = 1.26$, $P = 0.28$) during the conflict test, when compared to mCherry controls (gray bars, $n = 8$). 3V third ventricle, f fornix, sm stria medullaris, cc corpus callosum, MD mediodorsal thalamus, CA3 hippocampal CA3 subregion. Scale bars: 200 µm; inset scale bar: 25 µm. Two-way repeated-measures ANOVA followed by Bonferroni post hoc test. Data are shown as mean ± SEM. *$P < 0.05$. See also Supplementary Movie 9.

increased glutamate levels in the NAcSh has been correlated with reduced feeding behavior[45,46], and blockade of AMPA receptors in this same area increases food consumption[46,47]. Interestingly, a previous study has shown that PVT contains a subpopulation of neurons that express the glucose transporter Glut2 and project to the NAc to increase food-seeking behavior during hypoglycemia[15]. Thus, distinct subtypes of neurons projecting from the aPVT to the NAc may modulate food-seeking responses in opposite directions. aPVT$^{CRF}$ and Glut2-containing PVT neurons may work in

synchrony to regulate the opposing pressures of predation and starvation. This functional heterogeneity could help to explain why our nonselective pharmacological inactivation of aPVT neurons had no effect in food-seeking responses.

Our data showing that photoactivation of aPVT$^{CRF}$-NAc fibers evokes short-term facilitation in CINs suggest a potential role of CINs in mediating the food-seeking suppression observed in our experiments. Despite comprising <3% of all NAc neurons, CINs form a dense plexus of local innervation that can both

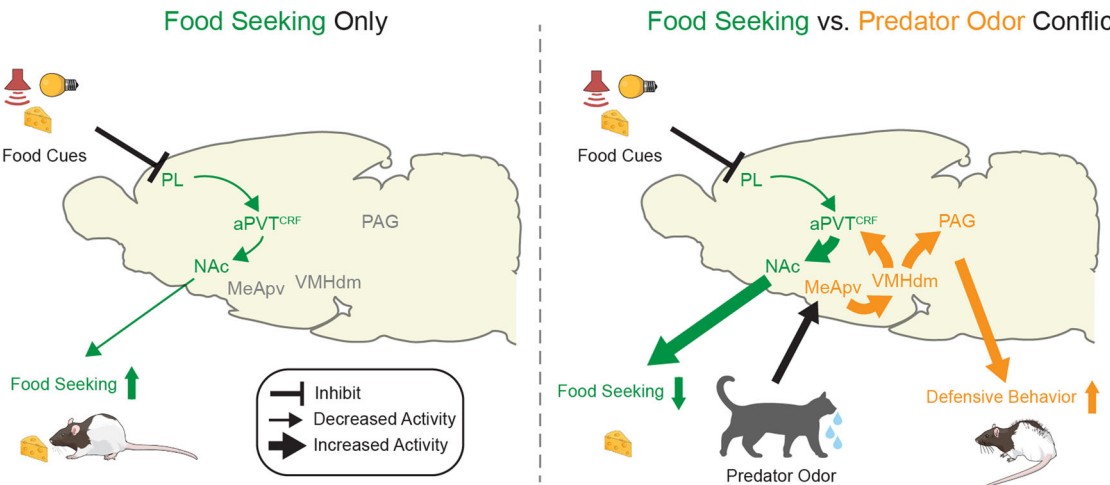

**Fig. 10 Schematic of our proposed model for the neural networks regulating predator-threat vs. food-seeking conflict.** (Left) During food-seeking behavior, food-associated cues inhibit the activity of PL glutamatergic neurons that project to aPVT[CRF], resulting in reduced activity in the aPVT[CRF]-NAc projection and consequently increased food-seeking responses[14]. (Right) During food-seeking vs. predator-threat conflict, cat odor activates MeApv neurons that project to VMHdm[50]. Subsequent activation of VMHdm neurons that project to PAG mediates defensive responses[51], whereas activation of VMHdm neurons that project to aPVT[CRF] suppresses food seeking (Supplementary Fig. 8a–c). Increased activity in the aPVT[CRF]-NAc pathway leads to target-dependent synaptic transmission in the NAc (Fig. 8), thereby reducing food-seeking behavior (Fig. 7).

modulate the firing rate of MSNs and directly control the release of dopamine from presynaptic terminals in the NAc, thereby regulating food-seeking motivation[23]. Accordingly, CINs either reduce or increase their firing rates in response to food cues when animals are more or less motivated to search food, respectively[48]. Moreover, activation of CINs in the NAcSh reduces food seeking, whereas inhibition increases it[49]. Our observation that aPVT[CRF] neurons exhibit either excitatory or inhibitory food-cue responses during the conflict suggests the existence of distinct subtypes of aPVT[CRF] neurons, which may have different anatomical targets. We speculate that the excitatory food-cue responses recorded during the conflict arise from aPVT[CRF]-NAc neurons, which provide a stop signal to accumbal neurons to suppress food seeking in the presence of potential threats. A recent study has shown that aPVT-NAc neurons receive inputs from PL and LH neurons, which, respectively, contribute to cued food seeking and feeding responses[14]. We have found that both PL and LH neurons send direct projections to aPVT[CRF] neurons, but how these pathways contribute to the balance between approaching food and avoiding predator threats should be the subject of further studies.

Exposure to cat odor activates the MeApv, the first telencephalic recipient of predator chemosignals following the accessory olfactory bulb[26]. MeApv neurons send dense projections to VMHdm, a pathway essential for predator odor recognition[50]. Furthermore, chemogenetic inactivation of VMHdm neurons attenuates freezing and risk-assessment behaviors in response to a predator[18], whereas photoactivation of these neurons induces a range of defensive behaviors that varies from avoidance, to freezing, to escape responses according to the intensity of laser stimulation[19], or the different VMHdm efferents activated[51]. Thus, it is possible that inactivation of collaterals from VMH-aPVT neurons to these two regions could mediate part of the reduced defensive responses observed in our study. However, our behavioral data showing that inhibition of VMH-aPVT neurons did not change rats' defensive behavior when the animals are exposed to cat odor alone suggest that such a possibility is unlikely. Our observation that inhibition of VMH-aPVT neurons reduced defensive behaviors but had no effect on food-seeking responses indicates that, in addition to VMH, other inputs to aPVT may also contribute to the food-

seeking suppression observed during the conflict. In the present study, a set of findings suggest that VMH may convey predator-related information to aPVT[CRF] neurons during the conflict: (i) aPVT[CRF] neurons receive inputs from VMH, (ii) VMH neurons make glutamatergic synapses onto aPVT-NAc neurons; (iii) exposure to cat odor activates VMH neurons that project to aPVT, (iv) inactivation of VMH-aPVT neurons reduces anti-predator defensive responses exclusively during the conflict, and (v) activating VMH-aPVT neurons is sufficient to mimic the cat odor-induced food-seeking suppression and the avoidance responses observed during the conflict.

Altogether, our results suggest that ascending inputs from VMH are necessary to transmit predator-threat signals to aPVT[CRF] neurons, which in turn modify synaptic transmission in downstream targets in the NAc that may serve to regulate foraging behavior under the risk of predation. Understanding the neural mechanisms that govern the opposing drives of approaching reward and avoiding threat can help to elucidate response selection and adaptive behaviors in humans.

## Methods

**Animals.** All experimental procedures were approved by the Center for Laboratory Animal Medicine and Care of The University of Texas Health Science Center at Houston. The National Institutes of Health guidelines for the care and use of laboratory animals were strictly followed in order to minimize any potential discomfort and suffering of the animals. Male and female Long–Evans hooded adult rats (Charles Rivers Laboratories) with 3–5 months of age and weighing 330–450 g at the time of the experiment were used. Animals were single housed and, after a 3-day acclimation period, handled and trained to press a lever for sucrose as described below. Rats were kept in a 12-h light/12-h dark cycle (light from 07:00 to 19:00) with water ad libitum. Animals were maintained on a restricted diet of 18 g per day of standard laboratory rat chow. Animals' weight was monitored weekly to make sure all animals maintained their weight under food restriction. During pre- and post-surgery phases, animals had ad libitum food access for a total of 7 days.

**Surgeries.** Rats were anesthetized with 5% isoflurane in an induction chamber. Animals were positioned in a stereotaxic frame (Kopf Instruments) and anesthesia was maintained with 2.5% isoflurane delivered through a facemask. A heating pad was positioned below the body of the animal and both temperature and respiration were monitored during the entire surgery. Animals received a subcutaneous injection of the local anesthetic bupivacaine (0.25%, 0.3 ml) at the incision site. Iodine and ethanol (70%) were alternately rubbed for the asepsis of the incision site. The surgery procedures varied according to the type of implantation/injection

(see below). The veterinary ointment was applied to the eyes to avoid dryness during the surgery. For injection-only surgeries, the incision was stitched after the injection by using a surgical suture (Nylon, 3-0). For implantation surgeries, the implants were fixed to the skull using C&B metabond (Parkell), ortho acrylic cement, and four to six anchoring screws. After surgery, animals received a subcutaneous injection of meloxicam (1 mg/kg) and a topic triple antibiotic was applied in the incision area.

*Cannula implantation.* For pharmacological inactivation experiments, a single guide cannula (26 gauge, 7 mm of length, Plastics One) was implanted aiming at the aPVT by using the following coordinates: AP −2.5 mm from bregma; mediolateral (ML) −1.83 mm from midline; dorsoventral (DV) −4.60 mm from the skull surface, at a 20° angle. A stainless-steel obturator (33 gauge) was inserted into the guide cannula to avoid obstruction until infusions were made.

*Viral vector injection.* Viral injections were performed using a microsyringe (SGE, 0.5 µl) with an injection rate of 0.05 µl/min plus an additional waiting time of 10 min to avoid backflow. The following coordinates from bregma were used: aPVT, −2.5 mm AP, −1.83 mm ML, and −5.6 mm DV at a 20° angle; NAc, +1.0 mm AP, ±2.3 mm ML, and −7.0 mm DV at an 11° angle; VMH, −2.5 mm AP, −1.7 mm ML, and −9.3 mm DV at a 10° angle. For chemogenetic inhibition, the Gi protein-coupled receptor hM4Di was expressed in target neurons using viral vectors. The hM4Di is a designer receptor exclusively activated by designer drugs (DREADDs) that can inhibit neuronal activity in the presence of CNO. To inhibit aPVT-NAc projections, 0.4 µl of a retrograde adeno-associated viral vector (AAV; pENN-AAVrg-hSyn.HI.eGFP-Cre, Addgene) was bilaterally injected into the NAc to express Cre recombinase in NAc inputs, followed by the injection of 0.5 µl of AAV-hSyn-DIO-hM4Di-mCherry (Addgene) or AAV-hSyn-DIO-mCherry (Addgene) was injected into the aPVT to Cre-dependently express either the hM4Di receptor or the control fluorescent protein mCherry specifically in NAc-projecting neurons. To inhibit VMH-aPVT projections, 0.5 µl of pENN-AAVrg-hSyn.HI.eGFP-Cre was injected into the aPVT to express Cre recombinase in aPVT inputs, whereas 0.4 µl of AAV-hSyn-DIO-hM4Di-mCherry or AAV-hSyn-DIO-mCherry was unilaterally injected into the VMH to Cre-dependently express either the hM4Di receptor or the control fluorescent protein mCherry in aPVT-projecting neurons. To minimize the spread of virus to other neighboring hypothalamic regions that also project to aPVT (e.g., Arc, DMH, and LH) and increase our chances of preferentially labeling VMH neurons, we used a smaller volume of virus and restricted our injection to one side of VMH only. For inhibition of aPVT[CRF] neurons, a viral vector expressing Cre recombinase under the control of rat CRF promoter (AAV-1/2-CRF-Cre-WPRE, $1 \times 10^{12}$ GP/ml, Genedetect) was used to specifically target CRF neurons. A 0.5 µl volume of an 1:1 viral mix containing either AAV-CRF-Cre plus AAV-hSyn-DIO-hM4Di-mCherry or AAV-CRF-Cre plus AAV-hSyn-DIO-mCherry was injected into the aPVT to express hM4Di receptor or the control fluorescent protein mCherry selectively in aPVT[CRF] neurons.

To optogenetically stimulate aPVT[CRF] neurons or aPVT[CRF]-NAc projections, the light-activated cation channel channelrhodopsin (ChR2) was selectively expressed in aPVT[CRF] neurons. A Cre-dependent eYFP protein was expressed in the same region to control for any nonspecific effects of viral infection or laser heating. A 0.5 µl volume of an 1:1 viral mix containing AAV-CRF-Cre plus either AAV-EF1a-DIO-hChR2(H134)-eYFP (UNC Vector Core) or AAV-EF1a-DIO-eYFP (UNC Vector Core) was injected into the aPVT to express ChR2 or eYFP, respectively. For anterograde tracing of aPVT[CRF] outputs, a glass pipette (outer diameter of ~40 µm) connected to a pressure injection device (Picospritzer, Parker Hannifin) was used to inject 0.5 µl volume of an 1:1 viral mix of AAV-CRF-Cre and AAV-EF1a-DIO-eYFP into the aPVT to selectively express fluorescent protein eYFP in the soma and axonal fibers.

Monosynaptic retrograde tracing of aPVT[CRF] neurons was carried out using a previously described method[52]. A glycoprotein-deleted pseudotyped rabies virus that can infect TVA-expressing neurons and trans-synaptically infect their presynaptic neurons in the presence of glycoprotein was used. To express mCherry-fused TVA receptors and glycoprotein selectively in aPVT[CRF] neurons, a 0.2 µl volume of 1:1:1 viral mix containing AAV-CRF-Cre, AAV-EF1a-FLEX-TVA-mCherry (UNC vector core), and AAV-CA-FLEX-RG (UNC vector core) was injected into the aPVT. Three weeks later, 0.5 µl of EnvA-dG-Rv-EGFP ($1.12 \times 10^9$ TU/ml, Salk Institute) was injected into the aPVT to infect the TVA-expressing neurons. Animals were perfused with potassium phosphate-buffered saline (KPBS) followed by 10% buffered formalin 1 week after the rabies viral injection for histology.

*Optogenetics.* For aPVT soma illumination, an optical fiber (0.22 NA, 200 nm core, Inper) was implanted aiming at the aPVT by using the following coordinates from bregma: −2.5 mm AP; −1.83 mm DV; −5.5 mm DV at a 20° angle. For the illumination of aPVT terminals, bilateral optical fibers were implanted aiming at the shell portion of the NAc by using the following coordinates from bregma: +1.0 mm AP, ±2.3 mm ML, −7.0 mm DV at an 11° angle.

*Single-unit recordings.* An array of 32 microwires (Bio-Signal) was implanted aiming at the aPVT by using the following coordinates from bregma: −2.5 mm AP;

−1.83 mm DV; −5.5 mm DV at a 20° angle. For photoidentification of aPVT[CRF] neurons, an optrode array (32 channels, 200 nm core, Bio-Signal) was implanted at the same coordinates described above. The ground wire was wrapped against a grounding screw previously anchored into the skull. Two insulated metal hooks were implanted bilaterally into the cement to allow firm attachment of the array connector to the cable during recording.

**Drugs.** Pharmacological inactivation of aPVT neurons was performed with the GABA$_A$ receptor agonist muscimol (BODIPY TMR-X conjugate; Thermo Fisher Scientific). A stainless-steel injector extending 1 mm past the cannula tip was connected to a 10 µl Hamilton syringe with a polyethylene (PE-20) tubing. An infusion machine (Model 11 plus, Harvard Apparatus) allowed the microinjection of muscimol (0.11 nmol/0.2 µl) over a 1-min time period 1 h before testing. After infusion, the injector was kept inside the cannula for 2 min to prevent backflow. For chemogenetic inhibition, CNO (Tocris) was used as a ligand for the inhibitory receptor hM4Di[53]. CNO was dissolved in 1% of dimethyl sulfoxide in sterile saline solution and intraperitoneally injected 1 h prior to the test (3 mg/kg).

**Behavioral tasks**

*Lever-press training.* Rats were placed in a plexiglass operant chamber (34 cm high × 25 cm wide × 23 cm deep, Med Associates, see schematic drawing in Fig. 5i) and trained to press a lever for sucrose on a fixed ratio of one pellet for each press. Next, animals were trained in a variable interval schedule of reinforcement that was gradually reduced across the days (one pellet every 15, 30, or 60 s) until the animals reached a minimum criterion of 10 presses/min. All sessions lasted 30 min and were performed on consecutive days. Sucrose pellet delivery, variable intervals, and session duration were controlled by an automated system (ANY-maze, Stoelting). Lever-press training lasted ~1 week, after which animals were assigned to surgery or cued food-seeking training. A small number of rats failed to reach the lever-press criteria and were excluded from the experiments (<3%).

*Cued food-seeking training.* Rats previously trained to press a lever for sucrose were trained to learn that each lever press in the presence of an audiovisual cue (tone: 3 kHz tone, 75 dB; light: yellow, 100 mA; 30 s duration) resulted in the delivery of a sucrose pellet in a nearby dish. While the light cue helps to direct the animal toward the lever during the beginning of the training phase, the tone assures that the animals would not miss any trial and provides the temporal precision required for single-unit recordings. After ~3 consecutive days of training (24 trials per day, pseudorandom intertrial interval of ~120 s, 60 min session), rats learned to discriminate the food-associated cue as indicated by a significant increase in press rate during the presence of the audiovisual cues, when compared to the 30 s immediately before the cue onset (cue-off, see Supplementary Fig. 1). The cued food-seeking training was completed when animals reached 50% of the discriminability index (presses during cue-on period minus presses during cue-off period divided by the total number of presses)[12].

*Cat odor collection and preparation.* Cat saliva was used as a source of cat odor because it is quantifiable, relatively easy to collect, and has been shown to induce robust defensive responses in rodents[24]. Cat saliva collection was performed in collaboration with a veterinary clinic in our city (https://www.snapus.org). A stripe of filter paper (0.5 cm × 6 cm) was gently introduced into the mouth of a cat and kept underneath its tongue for ~10 min while the animal was recovering from an incidental anesthesia. The saliva-soaked stripes were placed in a 50 ml falcon tube, transported back to the lab at 4 °C, and stored at −20 °C until the day of experimentation. The amount of saliva in each filter paper was estimated by weighing the identified filter paper stripes before and after the collection. Our pilot experiments indicated that ~100 µl of cat saliva is sufficient to trigger innate defensive responses in rats. Saliva was collected from cats of different sex, strains, and ages. Previous studies and our pilot tests indicate that odors collected from different cats (male vs. female, feral vs. domestic, young vs. adult) induce similar patterns of defensive responses[54]. The filter paper stripes impregnated with the cat saliva were placed in a half-sphere metal mesh, which was anchored to the wall of the behavior chamber nearby the food dish.

*Approach-avoidance conflict test.* After completing the cued food-seeking training, rats were placed into a plexiglass rectangular arena (40 cm high × 60 cm wide × 26 cm deep, Med Associates, see schematic drawing in Fig. 1a) and re-trained to press for sucrose in the presence of the same audiovisual cues until they reached 50% of discriminability index. The arena consisted of a hidden area (40 cm high × 20 cm wide ×26 cm deep) separated from an open area by a plexiglass division. An 8-cm slot located in the center of the division enabled the animal to transition from both sides of the arena. For behavioral quantification, the open area was subdivided into a center area and a food area (40 cm high × 12 cm wide × 26 cm deep), the latter being equipped with a lever, a dish, and an external feeder similar to a food-seeking operant chamber. During the baseline, animals were placed in the arena in the presence of a neutral odor (filter paper stripes soaked with distilled water) for 10 min without any other stimuli. Next, 12 audiovisual cues (30 s duration, pseudorandom intertrial intervals in a range 30–90 s) were presented during an additional 20 min. The total duration of the session last 30 min. Each

lever press in the presence of the cues resulted in the delivery of a sucrose pellet into the dish. During the conflict test, the following day, rats were placed in the arena in the presence of the cat odor only (filter paper stripes soaked with cat saliva) for 10 min. Next, the audiovisual cues were concomitantly presented to the animals using the same intervals of the baseline. Avoidance responses were characterized by the time spent in the hidden area of the arena. Freezing responses were characterized by the complete absence of movements except those needed for respiration. Head-out responses were characterized by a body stretching movement to peep out toward the odor mesh while in the hidden area, and were used as a measure of risk-assessment behavior.

For the single-unit recording experiments, animals received an extra day of cued food-seeking training in which the audiovisual cue was turned off after each lever press, thereby reducing the rat's response to a single press and dish entry per cue. This enabled us to correlate each food-seeking event with the firing rate of aPVT neurons. To precisely track the activity of the same aPVT neuron in response to different stimuli, rats were exposed to a single recording session separated into three different phases: (i) a food-seeking phase (12 audiovisual cues in the presence of a neutral odor, 10 min duration), (ii) a cat odor phase (10 min of cat odor exposure, no audiovisual cues), and (iii) a conflict test (24 audiovisual cues in the present of cat odor, 30 min duration). After each animal, the arena was cleaned thoroughly with 70% ethanol solution, except the odor area.

*Cued food-seeking test*. After completing the cued food-seeking training, rats were returned to the same chamber and tested for their baseline pressing rate with 16 audiovisual cues (pseudorandom intertrial intervals, 32 min session). The next day, a laser was delivered during two consecutive cues (laser on at cue onset) followed by two consecutive cues without laser (laser off). During chemogenetic experiments, CNO was administered 1 h before the test session. After each animal, the arena was cleaned thoroughly with 70% ethanol solution.

*Real-time place preference test*. The place preference apparatus consisted of an acrylic chamber (40 cm high × 62 cm wide × 31 cm deep) equally divided into two different compartments connected by an entry (7 cm wide). One compartment was composed of a hollow-dotted floor with white walls (laser-off side), whereas the other was composed of a flat-striped floor with black walls (laser-on side). Rats were placed in the laser-off side and the blue laser was activated to illuminate either aPVT$^{CRF}$ neurons or their terminals in the NAc every time the animal crossed to the laser-on side of the chamber. The laser remained on for a maximum of 20 s or until the animal crossed back to the laser-off side. The percentage of time spent and the distance traveled in each side of the chamber were automatically assessed (ANY-maze) during the 10 min of the session. After each animal, the chamber was cleaned thoroughly with 70% ethanol solution.

*Shuttle food-seeking test*. Rats previously trained to press a lever for sucrose in the presence of an audiovisual cue were trained for 3 additional days in the shuttle food-seeking test (see schematic drawing in Supplementary Figure 8). The same arena used for the approach-avoidance conflict test was modified to contain two levers, two feeders, two food dishes, two speakers, and two light cues positioned in opposite walls of the chamber. Rats were trained to alternate between both sides of the arena and receive sucrose pellets after they press the lever in the side of chamber signaled by the audiovisual cues. During the test session, the laser was activated to photo-activate VMH-aPVT projections each time the animals entered the food area in the presence of the audiovisual cues as an attempt to mimic the cat odor conflict. Two consecutive laser-off trials were followed by two laser-on trials in a total of ten trials.

**In vivo optogenetic stimulation**. Unilateral or bilateral optical cables (200 μm core, 0.37 NA, 2.5 mm ceramic ferrule, Inper) were connected to a blue laser (diode-pumped solid-state, 473 nm, 150 mW output, OptoEngine) by using a patch cord (200 μm, 0.22 NA, FC/PC connector, Inper) through a single or dual rotary joint (200 μm core, Doric lenses). During the stimulation, the optical cables were coupled to the previously implanted optical fibers by using a ceramic sleeve (2.5 mm, Precision Fiber Products). An optogenetic interface (Ami-2, Stoelting) and an electrical stimulator (Master-9, A.M.P. Instruments) were used to control the onset of the laser, pulse duration, and frequency. The used parameters of photoactivation were 20 Hz and 5 ms pulse width for illumination of either aPVT$^{CRF}$ somata or aPVT$^{CRF}$-NAc projections, and 20 Hz and 20 ms pulse width for illumination of VMH-aPVT projections. The power density estimated at the tip of the optical fiber was 7–10 mW for illumination of aPVT$^{CRF}$ somata, 15 mW for aPVT$^{CRF}$-NAc projections and 20 mW for VMH-aPVT projections (PM-100D, Power Energy Meter, Thor Labs).

**In vivo single-unit electrophysiology**. A 64-channel neuronal data acquisition system (Omniplex, Plexon) integrated with a high-resolution video-tracking system (Cineplex, Plexon) was used for electrophysiological recordings from freely behaving animals. Both videos and neuronal recordings were combined within the same file, thereby facilitating the correlation of behavior with neuronal activity. The system was connected to the head-mounted array by using a digital headstage cable (32 channels, Plexon), a motorized carrousel commutator (Plexon), and a digital headstage processor (Plexon). Rats were habituated to the headstage cable

daily for ~1 week before the beginning of the experiments. Extracellular waveforms exceeding a voltage threshold were band-pass filtered (500–5000 Hz), digitized at 40 kHz, and stored onto disk. Automated processing was performed using a valley-seeking scan algorithm and then visually evaluated using sort quality metrics (Offline Sorter, Plexon). Single units were selected based on three principal components and waveform features such as valley-to-peak and amplitude measurements. Commercial software (NeuroExplorer, NEXT Technologies) and MATLAB scripts were used to calculate the spontaneous firing rate, food-cue responses, lever-press responses, and dish-entry responses. The spontaneous firing rate was calculated by comparing the frequency of spike trains during the last 30 s of the food-seeking phase, cat odor phase, or conflict test against the 30 s prior to each session. Food-cue, lever-press, and dish-entry responses were calculated as Z-scores normalized to 20 pre-cue bins of 300 ms. Neurons showing a Z-score >2.58 ($P < 0.01$) during the first two bins following the onset of the food cue, lever press, or dish entry were classified as excitatory responses, whereas neurons showing a Z-score < −1.96 ($P < 0.05$) during the same first two bins were classified as inhibitory responses. At the end of the recording sessions, a microlesion was made by passing an anodal current (0.3 mA for 15 s) through the active wires to deposit iron in the tissue. After perfusion, brains were extracted from the skull and stored in a 30% sucrose/6% ferrocyanide solution to stain the iron deposits.

For correlation analyses, we applied a Z-score criterion to compute the mean and standard deviations of firing rates (spikes/s) considering 6 s before the stimulus onset for normalization that was above or below the mean across time. This criterion is a dimensionless quantity derived by subtracting the data mean from an individual firing rate and then dividing the difference by the standard deviation. The firing rates were computed per food-cue presentation for each cell within 300 ms bins. The binned firing rates calculated from the stimulus onset until the animal pressed the lever within 3 s were averaged to represent the response of the neuron for that trial. If the latency to press was longer than 3 s, then the mean firing rate was computed using the first 3 s after food-cue onset. The Pearson's correlation coefficient was used to measure the relationship between mean firing rate and latency to press (latencies within 0–30 s). The correlation was computed using a vector containing all neuronal responses and a vector containing the latency to press across all the food-cue presentations. Values were divided into excitatory (Z-score >0) vs. inhibitory (Z-score <0) food-cue responses and presented in different scatter plots.

**In vivo photoidentification of aPVT$^{CRF}$ neurons**. During photoidentification of aPVT$^{CRF}$ neurons, we recorded from rats expressing ChR2 in aPVT$^{CRF}$ neurons. An optical cable was connected to the blue laser by using a patch cord through a single rotary joint. The optical cable was attached to the headstage cable and coupled to the previously implanted optical fiber by using a ceramic sleeve. At the end of the behavioral session, 10 trains of 10-s light pulses (5 ms pulse width, 5 Hz, 10 mW) were delivered with the blue laser controlled by a Master-9 programmable pulse stimulator, which also sent flags to the data acquisition system to mark the time of the laser events. Neurons were considered to be responsive to photo-stimulation if they showed a significant increase in firing rate above baseline (20 ms, Z-score >3.29, $P < 0.001$) within the 12 ms after laser onset, as previously described[38].

**In vitro electrophysiology and optogenetics**

*Slice preparation*. For recordings in the NAc, we used acute brain slices from rats expressing ChR2-eYFP in aPVT$^{CRF}$ neurons. For PVT recordings, we used slices from rats previously injected with the retrograde tracer CTB into the NAc and ChR2 into the VMH. Brain slices were prepared as described previously[55]. Briefly, animals were deeply anesthetized using isoflurane and transcardially perfused using an ice-cold $N$-methyl-D-glutamine (NMDG)-based solution saturated with 95% O2/5% CO2 consisting of the following (in mM): 92 NMDG, 2.5 KCl, 1.25 NaH$_2$PO$_4$, 10 MgSO$_4$, 0.5 CaCl$_2$, 30 NaHCO$_3$, 20 glucose, 20 HEPES, 2 thiourea, 5 Na ascorbate, and 3 Na pyruvate. Brains were removed from the skull and coronal slices (300 μm) were cut in ice-cold NMDG-based solution using a vibratome (VT1200 S, Leica). Slices were then held in NMDG-based solution maintained at 35 °C for ~12 min before being transferred to a modified artificial cerebrospinal fluid (ACSF) held at room temperature, consisting of the following (in mM): 92 NaCl, 2.5 KCl, 1.25 NaH$_2$PO$_4$, 2 MgSO$_4$, 2 CaCl$_2$, 30 NaHCO$_3$, 25 glucose, 20 HEPES, 2 thiourea, 5 Na ascorbate, and 3 Na pyruvate.

*Electrophysiology and optogenetics*. Recordings were performed in a chamber perfused with ACSF consisting of the following (in mM): 126 NaCl, 2.5 KCl, 1.25 NaH$_2$PO$_4$, 2 MgCl$_2$, 2 CaCl$_2$, 26 NaHCO$_3$, and 10 glucose, held at 32–34 °C using an inline heater (TC-324B, Warner Instruments). Cells were visualized via infrared differential interference contrast under an Olympus BX51WI microscope equipped with a Dage-MTI IR-1000 camera. All recordings were made using borosilicate glass pipettes (3–6 MΩ) filled with intracellular solution containing (in mM): 133 K gluconate, 1 KCl, 2 MgCl$_2$, 0.16 CaCl$_2$, 10 HEPES, 0.5 EGTA, 2 Mg-ATP, and 0.4 Na-GTP (adjusted to 290 mOsm and pH 7.3). For NAc recordings, we used epi-fluorescence to target areas with strong eYFP expression in aPVT$^{CRF}$ synaptic afferents. Putative CINs were differentiated from neighboring MSNs by their larger cell bodies and the presence of rebound firing following a hyperpolarizing current step[56]. Pair of MSNs and CINs located within 50 μm and at similar depth were

recorded in sequence[57]. For PVT recordings, neurons labeled retrogradely from the NAc were targeted using epifluorescence. Excitatory postsynaptic responses were evoked using blue light pulses (1 ms) generated by an LED light source (UHP-T-450-EP, Prizmatix), and delivered through a ×60, 0.9 NA water-immersion objective (Olympus).

*Data acquisition*. Recordings were acquired using a MultiClamp 700B amplifier (Molecular Devices), filtered at 3–10 kHz, and digitized at 20 kHz with a 16-bit analog-to-digital converter (Digidata 1440A; Molecular Devices). Data were acquired using Clampex 10.3 (Molecular Devices) and analyzed using custom macros written in IGOR Pro (Wavemetrics).

**Histology**. Animals were transcardially perfused with KPBS followed by 10% buffered formalin. Brains were processed for histology as previously described[58]. Only rats with the spread of fluorescent muscimol, the presence of eYFP or mCherry labeling, and the track of the electrode wires or optical fiber tips located exclusively in the target area were included in the statistical analyses.

**Immunohistochemistry**. Rats were perfused with KPBS followed by 10% buffered formalin 90 min after the odor exposure. Brains were removed from the skull, transferred to a 20% sucrose solution in KPBS for 24 h, and stored in a 30% sucrose solution in KPBS for another 24 h. Next, coronal brain sections (40 μm thick) were cut in a cryostat (CM1860, Leica), blocked in 20% normal goat serum, and 0.3% Triton X-100 in KPBS at room temperature for 1 h. For brightfield cFos immunohistochemistry, brain sections were incubated with anti-cFos serum raised in rabbit (1:15,000; EMD Millipore) at 4 °C for 48 h. After sections were washed in KPBS five times, sections were incubated with Biotinylated goat anti-rabbit IgG antibody (1:200, Vector Labs) for 2 h, followed by incubation in ABC Kit (1:200, Vector Labs) for 90 min and DAB-Ni solution (Vector Labs) for 15–20 min. Sections were washed with KPBS, mounted in Superfrost Plus slides, and cover-slipped with mounting medium (Vectashield, Vector Labs).

For fluorescent cFos immunohistochemistry, brain sections from rats expressing the fluorescent reporter eYFP in either aPVT$^{CRF}$ neurons (previously infused with AAV-CRF-Cre plus AAV-DIO-eYFP into the aPVT) or in afferent neurons of aPVT (previously infused with retrograde AAV-Cre-eYFP into the aPVT) were blocked in 20% normal goat serum and 0.3% Triton X-100 in KPBS at room temperature for 1 h, and then incubated with anti-cFos serum raised in rabbit (1:1000; EMD Millipore) at 4 °C for 48 h. After sections were washed in KPBS five times, sections were incubated with anti-rabbit secondary-antibody Alexa Fluor 594 raised in goat (1:200, Abcam) for 2 h, washed with KPBS, mounted in Superfrost Plus slides, and cover-slipped with anti-fading mounting medium with 4′,6-diamidino-2-phenylindole (DAPI) (Vector Labs). For fluorescent labeling of aPVT$^{CRF}$ projections (Fig. 6), brain sections were preincubated in a blocking solution (5% donkey serum, 0.3% Triton X-100, and 0.1% sodium azide in 0.1 M PBS) at room temperature for 1 h, and then incubated with anti-GFP serum raised in rabbit (1:1000; Invitrogen) at room temperature overnight. Sections were rinsed in KPBS three times, incubated in anti-rabbit secondary-antibody Cy2 raised in donkey (1:500, Jackson ImmunoResearch) for 2 h, washed in KPBS, mounted in gelatin-coated slides, and cover-slipped with anti-fading mounting medium (Fluoromount-G, Thermo Fisher Scientific).

**In situ hybridization**. Single-molecule fluorescent in situ hybridization (RNAscope Multiplex Fluorescent Detection Kit v2, Advanced Cell Diagnostics) was used following the manufacturer's protocol for fixed-frozen brains sample. Brain samples were sectioned at a thickness of 20 μm in a cryostat (CM1860, Leica). Sections were collected onto superfrost plus slides (Fisher Scientific) and transferred to a −80 °C freezer. To prepare for the assay, brain sections were serially dehydrated with EtOH (50, 75, and 100%, each for 5 min) and then incubated in hydrogen peroxide for 10 min. Target retrieval was performed with RNAscope target retrieval reagents at 99 °C for 5 min. The sections were then pretreated with Protease III (RNAScope) for 40 min at 40 °C. RNAscope probes (Advanced Cell Diagnostics) for eYFP (cat. 312131) and CRF (cat. 318931-C3) were hybridized at 40 °C for 2 h, serially amplified, and revealed with horseradish peroxidase, Opal Dye/TSA Plus fluorophore (Akoya Biosciences), and horseradish peroxidase blocker. Sections were cover-slipped with anti-fading mounting medium with DAPI (Vectashield, Vector Labs) and kept in the refrigerator. Images were generated by using a confocal laser scanning (Zeiss 800) at ×40 or ×63 oil immersion objective with the appropriate filter sets. Colocalization of eYFP messenger RNA (mRNA) (green, Opal 520) and CRF mRNA (red, Opal 620) was manually counted by an experimenter measuring either the percentage of eYFP-positive neurons in aPVT that were also labeled with CRF, or the percentage of CRF-positive neurons in aPVT that were also labeled with eYFP.

**Microscopy and image analysis**. Images were generated by using both a confocal laser scanning (Zeiss 810) and a microscope (Nikon, Eclipse NiE Fully Motorized Upright Microscope) equipped with a fluorescent lamp (X-Cite, 120 LED) and a digital camera (Andor Zyla 4.2 PLUS sCMOS). Counts of cFos-positive neurons were performed at ×20 magnification. Cells were considered positive for cFos-like immunoreactivity if the nucleus was the appropriate size (area ranging from 100 to 500 mm²) and shape (at least 30% of circularity), and presented a black/brown color (for brightfield) or a red fluorescent color (for fluorescence) different from the background. cFos-positive cells were automatically counted (NIE Element Software) and averaged for both hemispheres at 2–3 different AP levels of each area: for aPVT, from −1.3 to −2.3 mm from bregma; for posterior PVT, from −2.8 to −3.6 mm from bregma; for MeApv and MeApd, from −2.8 to 3.3 mm from bregma; for VMHdm and VMHvl, from −2.3 to −3.1 mm from bregma, for PL and IL, from +3.7 to +2.2 mm, and for PAGdl and PAGvl, from −6.3 to −8.0 mm from bregma. The density of cFos-positive cells (cells per 0.1 mm²) was calculated by dividing the number of cFos-positive cells by the total area of each region. cFos colocalization with eYFP fluorescence was manually quantified by an experimenter blind to experimental groups by measuring the percentage of cFos-positive neurons that were also labeled with eYFP, or the percentage of eYFP-positive neurons that were also labeled with cFos.

**Statistics and reproducibility**. Rats were recorded with digital video cameras (Logitech C920) and behavioral indices were measured using automated video-tracking system (ANY-maze). Manual counting was performed by an experimenter blind to the experimental groups to quantify the percentage of head-out time. Presses per minute were calculated by measuring the number of presses during the 30 s cue multiplied by two. All graphics and numerical values reported in the figures are presented as mean ± s.e.m. All experiments were performed in multiple batches of replication at least three times per experiment. Data were then combined after histological verification of implant position and/or viral vector expression to generate the final sample size of each group. Statistical significance was determined with two-tailed paired or unpaired Student's *t* test, repeated-measures analysis of variance followed by Bonferroni post hoc comparisons (Prism 7), and *Z* test or Fisher's exact test, as indicated for each experiment. The sample size was based on estimations by power analysis with a level of significance of 0.05 and a power of 0.9.

**Reporting summary**. Further information on research design is available in the Nature Research Reporting Summary linked to this article.

## Data availability

All the data that support the findings presented in this study are available from the corresponding author on reasonable request.

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

## Acknowledgements

We thank Dr. Nicholas Justice (UTHealth) for discussions during the initial stage of this project, Dr. De-Pei Li (University of Missouri) for providing us the CRF-Cre viral construct, the SNAP Spay-Neuter & Animal Wellness Clinic at Houston for helping us with the cat saliva collection, and Ana Terzian, Leah Olivo, Maria Rasheed, and Sharon Gordon for their technical assistance. We also thank current and former members of the Do Monte and Quirk Labs for their valuable comments on the manuscript, and the Mind the Graph team for creating the schematic drawings presented in Figs. 1a, 2a, 5i, m 7b, f, and 10 and Supplementary Figures 1i, 3a, 6f–g, and 8b–d. This work was supported by a CIHR grant MOP-89758 to G.J.K. and an NIH grant R00-MH105549, an NIH grant R01-MH120136, a Brain & Behavior Research Foundation grant (NARSAD Young Investigator), and a Rising STARs Award from UT System to F.H.D.-M.

## Author contributions

D.S.E., and X.O.Z. performed and analyzed the behavioral, immunohistochemical, in situ hybridization, retrograde viral tracing, chemogenetic, and optogenetic experiments. J.J.O. and M.B. designed, performed, and analyzed the in vitro recordings. D.S.E., J.A.F.-L., and F.H.D.-M. collected and analyzed the single-unit recording data. S.L. and G.J.K. performed and analyzed the anterograde viral tracing. D.S.E. and F.H.D.-M. designed the study, interpreted the data, and prepared the manuscript with comments from all the co-authors.

## Competing interests

The authors declare no competing interests.
