## [Peer Review File · Nature Communications]

Reviewers' Comments:

Reviewer #1:

Remarks to the Author:

The authors make an important contribution to the field by 1) showing that CRF+ neurons in PVT that project to NAc are both necessary and sufficient to interrupt ongoing appetitive behaviors and favor defensive responses to stimuli, 2) showing that these neurons are entrained to food seeking cues and that these responses are greater in the presence of a competing threatening stimulus, and 3) showing that activation of these neurons is aversive. They also perform useful rabies tracing of inputs to CRF+ PVT neurons that overlap with the known inputs deriving from non-cell type specific tracing studies, and show evidence from in vitro ChR2-dependent circuit mapping for a di-synaptic VMH-PVT-NAc circuit.

These findings are important because they give a circuit basis to the predicted role of midline thalamic nuclei in conveying information about innate threat cues that derive from the medial hypothalamic and brainstem defense system. The innovation here is that the authors have identified the specific cell type in PVT and forebrain target by which this feedback system suppresses learned appetitive behaviors.

Overall the work is clearly presented and appears to be carefully carried out. The work is very comprehensive and represents a major and complete characterization of the connections investigated – a highly laudable accomplishment. Proper controls are done for optogenetic experiments, for example, with neutral fluorophores and sufficient animals are used throughout to ensure reasonable statistically testing.

Comment:

1. I found it very difficult to understand precisely what stimuli (type, time, frequency, ITI) and options the animals were being presented during the baseline, food only, odor only, and conflict phases, even when going through the Methods in detail. Readability would be significantly enhanced if this information were diagrammatically presented in more detail at the beginning of Figure 1. Following on this point, it was not clear to me to precisely which stimuli the single unit recording was aligned – 'Food cues'?

2. There is no discussion of the time course of the Food cue entrained neural activity signatures (Figure 2; both increased and decreased activity). The unit in Figure 2f, for example, has a very slow kinetics. Does this correlate (better) with other features of the cue or cue train or with the behavior of the animal? In general, there seems to be a fast phasic response and then a longer (ca. 3 s) persistent response. Here it would be good to show the length of the food cue and any relevant trial-by-behaviors directly on the graphs. Also, in Figure 2d,i it appears that there is a decrease in activity precisely 3 s before the food cue onset. What is this suppression entrained to? Why is this not discussed? The effect appears not to be evident in Figure 2e,g,j,l. Why?

3. It is not discussed why VMH-PVT projection inactivation (Figure 8h-m) suppresses defensive behaviors, but doesn't increase food approach or lever pressing, while PVT or PVT-NAc inactivation (e.g. Figure 5b-g) does both. Could this be explained by the suppression of VMH collaterals to downstream brainstem targets that produce the defensive responses, rather than via PVT projections. This discrepancy needs to be discussed.

4. Given the design of the box where the odor is right next to the food hopper any manipulation that increases defensive behaviors should also decrease lever pressing for food – so how is it possible that a manipulation affects only defensive and not food responding? If you had the odor and food hopper at opposite ends of the cage, for example, would they still interfere? In other words, a circuit manipulation that decreases defense would appear to have the same effect as a manipulation that

increases food seeking. How is this confound addressed in the study?

Reviewer #2:

Remarks to the Author:

This is an excellent study and well written manuscript by Engelke and colleagues. The authors provide very interesting findings on the neural coding of anterior paraventricular thalamus (aPVT) neurons of a rewarding, an aversive cue, and during a conflict where both cues are presented. The study also includes multiple causal manipulations (pharmacologic, optogenetic and chemogenetic) of the aPVT and of pathway selective neurons of this brain region. I have to say that it has been a pleasure to read and evaluate this study, and I only have minor suggestions.

My first suggestion is to tone down the conclusion about the functional role of the divergent connectivity between the MSN and CIN neurons of the NAc. Indeed, the authors did not record the synaptic strength or plasticity of this connection. I therefore suggest to change or remove "by mediating target-dependent synaptic transmission in the NAc" in the abstract and the result/discussion section.

As the main techniques is in vivo electrophysiological recordings, and although I trust recordings quality, showing examples raw traces of multiple channels, and some clusters in Figure 2a or in a supplementary figure would be beneficial.

To avoid misunderstanding please replace aPVT-projecting VMH neurons with VMH-aPVT neurons.

The authors provide a large number of long videos. To increase the value of this supplementary files, I would advise the authors to show all of them at 16x speed all the time rather than changing speed in the middle of videos and at different speeds across videos. Also, it would greatly increase the impact of the video to show them simultaneously rather than one after the other: mCherry on the top, DREADDs on the bottom – same for Chr/eYFP etc.. Also, please keep the experimental group of the rat the whole time on the video. Finally, this is optional, but a great addition would be to add the track of the animal on the video in red.

Figure 1

- k-o: please specify the y axis titles : is it raw numbers of cells ? or cells/mm² (which it should be)
- b: the color code seems to be black for neutral odor and green for cat odor – however, these are mixed up in c-h and k-o. Please make the neutral odor histograms black, as well as the dots, for the cat odor, keep the histograms green (ideally, filling 50% transparent and outline 100% green) and the dots with 100% green outline and white filling.

Also, in a, the neutral odor is grey – please make it black

- b and j: rather than rectangles surrounding the data, the authors could consider a thick black or green line on top of the graphs, and right below 'Neutral odor' and 'Cat odor'.

- same comments apply to Figure S1 and all figures.

Figure 2

- it remains unclear why the authors do not represent the same analysis for the cat odor than for the food-seeking (neurons responding to cat odor before vs. same neurons during conflict, as well as neurons responding during conflict vs. same neurons before conflict). I understand the time locked response with the odor is more difficult to define, however, they characterized 11% of the recorded neurons as 'cat odor responsive'. Their response during conflict should be assessed, at least in a supplementary figure

- c: It is written in the legend, but adding a legend on the figure that red is excitation and blue inhibition would be helpful.

Figure 2c/4j: why did the authors choose different representations for the proportion of neurons? Please be consistent. Moreover, statistical comparisons of these proportions would be very interesting.

Figure 8: part of the text has a different font than the rest of the figure.

In the discussion, the authors could consider citing the other electrophysiological studies during conflict of positive and negative cues such as Burgos-Robles et al. 2017 (PMID: 28436980).

Reviewer #3:

Remarks to the Author:

In their manuscript entitled "A hypothalamic-thalamostriatal circuit that controls approach-avoidance conflict" Engelke and colleagues use an ethologically relevant behavioral model to investigate how the aPVT – a structure that the corresponding author has previously shown to control both reward and defensive behaviors – arbitrates behavioral decisions amid motivational conflicts. Specifically, first, they trained rats in an instrumental task to obtain food reward then subsequently challenge these with the odor of a known predator, cat odor. In response to the motivational conflict between reward availability and cat odor, rats displayed defensive behaviors such as freezing and avoidance, and reduced their reward seeking (number of presses per minute, latency to press, etc.). Using this conflict task, the authors then used cFos mapping to identify regions of the brain selectively recruited by the predator odor. Among these they identified the MeA the VMH and the aPVT. Importantly, using modern neuroscience tools such as in vivo recordings from photo-tagged neurons, chemogenetics and optogenetics manipulations the authors identified a subpopulation of aPVT neurons that expresses CRF and are selectively recruited during the cat odor presentation. The authors identified these neurons as projecting to the NAc, and subsequently demonstrated that inhibition of aPVT-NAc communication or aPVTCRF somata attenuates defensive behavior and promotes food seeking amid motivational conflicts.

Overall, this is a very elegant study that uses an innovative and ethologically relevant behavioral model and defines a brain circuit that participates in guiding behavior decisions amid motivational conflict. In addition, the authors also demonstrate that subpopulations of PVT neurons modulate their responses to reward-predicting cues depending on the absence or presence of a motivational conflict. This particular finding builds on the notion that PVT neurons undergo robust state-dependent modulation of cue responses. In addition, the study presents a timely assessment of the role of PVT circuits in arbitrating amid conflicts. As such, I consider the study to be pioneering and highly important to the field. In short, I believe that the study will interest the broad readership of Nature Communications.

While I do not consider that further experimental work will be strictly necessary, below I raised several points that require clarification from the authors:

1) The observation that inhibition of the aPVT doesn't abolish defensive responses to 'Cat odor only' is very interesting. This observation seems to be in line with the notion that the PVT is not required for acute fear responses (for example during fear conditioning) but rather mediates fear responses to conditioned stimuli (fear memory retrieval). However, defensive responses during conflict are dependent on the aPVT. The authors should offer more conceptual insight into why they think that is. Such insight could help to advance important questions of the field.

2) Using cFos, the authors conclude that aPVTCRF neurons are activated during the conflict (cat odor). Interestingly, while in photo-tagged aPVTCRF neurons they also observed an increase in the percentage of neurons modulated by the food cue during the conflict compared to before the conflict, the more prominent effect observed here is in neurons that are inhibited by the food cue. This particular finding seems to be at odds with the authors' conclusions about the role of CRF neurons in

the PVT. Can the authors clarify this issue?

3) The observation that unlike whole aPVT silencing, exclusively silencing aPVT-NAc neurons rescues food pressing behavior during the conflict suggests that PVT circuits controlling food and aversive-related behaviors are somewhat segregated. The authors should offer additional insight on this point.

4) Related to the previous point, the authors identify the VMH as providing critical input to aPVTCRF neurons during conflict. However, whereas silencing aPVTCRF neurons diminishes defensive behaviors and rescues food seeking, silencing of aPVT-projecting VMH neurons attenuates defensive behaviors but do not rescue food seeking. This result is consistent with segregation of the aPVT circuits that promote defensive behavior and suppress reward seeking during conflict. There is a sense that heterogeneous populations of PVT neurons guide behavior during conflict. The authors should address these seemingly discrepant observations.

5) Also, the authors show that stimulation of aPVTCRF neurons mimics the defensive (avoidance) responses and attenuated food seeking observed during conflict. But what about freezing behavior? Inactivation of aPVT-NAc neurons attenuates freezing (Figure 3).

REVIEWER COMMENTS

Reviewer #1 (Remarks to the Author):

1. I found it very difficult to understand precisely what stimuli (type, time, frequency, ITI) and options the animals were being presented during the baseline, food only, odor only, and conflict phases, even when going through the Methods in detail. Readability would be significantly enhanced if this information were diagrammatically presented in more detail at the beginning of Figure 1. Following on this point, it was not clear to me to precisely which stimuli the single unit recording was aligned – ‘Food cues’?

To address the reviewer concern about the time, duration and interval of the different stimuli during our conflict test, we have improved the schematic drawings in both Fig. 1 and Fig. 2. We have also modified the Methods section to include more details. It now reads:

“During the baseline, animals were placed in the arena in the presence of a neutral odor (filter paper stripes soaked with distilled water) for 10 min without any other stimuli, then, 12 audiovisual cues (30 s duration, pseudorandom inter-trial intervals in a range of 30s to 90s) were presented during an additional 20 min, being the total time of the test 30 min. Each lever press in the presence of the cues resulted in the delivery of a sucrose pellet into the dish. During the conflict test in the next day, rats were placed in the arena in the presence of the cat odor only (filter paper stripes soaked with cat saliva) for 10 min. Then, the audiovisual cues were concomitantly presented to the animals using the same intervals of the baseline”

In addition, to make it clear to which stimuli the firing rate of the cells were time-locked to, in Fig.2 and Fig 4., we have introduced an arrow close to the word “food cues” on the top of the raster plots and PSTHs to indicate that time zero corresponds to the cue onset and emphasized that in the figure legends too. The same modifications were done for Supp. Fig 4 and Supp Fig. 5, when the firing rates were aligned to lever presses or dish entries.

2. There is no discussion of the time course of the Food cue entrained neural activity signatures (Figure 2; both increased and decreased activity). The unit in Figure 2f, for example, has a very slow kinetics. Does this correlate (better) with other features of the cue or cue train or with the behavior of the animal? In general, there seems to be a fast phasic response and then a longer (ca. 3 s) persistent response. Here it would be good to show the length of the food cue and any relevant trial-by-behaviors directly on the graphs.

We want to thank the reviewers for the insightful observation. We have addressed this point now by making some additional analyses and creating a new supplementary figure to demonstrate the percentage of sustained vs. fast tone responsive cells (See Supp. Fig. 5). We have added some raster plots/PSTHs examples of sustained vs. fast food-cue responses (Fig. 2 and Supp. Fig. 5, respectively). In addition, we have performed additional trial-by-trial analyses to investigate whether the firing rate of the food cue responses (fast vs. sustained) correlates with the latency to press the lever, but we found no correlation for either excitatory or inhibitory food-cue responses (Supp. Fig. 5).

As recommended by the reviewer, we have added the length of the food cue in the schematic drawings, included colored markers for cue onset, bar presses, and dish entries in each one of

the raster plots to indicate the different behaviors of the animals during each trial, and created new heatmaps to show the firing rate of all the cells aligned to bar presses and dish entries (see Fig. 2, Fig. 4, Supp. Fig. 4, and Supp Fig. 5). In addition, we have performed some analyses to time-lock the firing rate of aPVT neurons to other behaviors (e.g., onset of freezing, onset of avoidance), but we couldn't observe any significant differences (not shown).

Also, in Figure 2d,i it appears that there is a decrease in activity precisely 3 s before the food cue onset. What is this suppression entrained to? Why is this not discussed? The effect appears not to be evident in Figure 2e,g,j,l. Why?

We appreciate the sharp eyes of the reviewer on this point. We are sorry for the misunderstanding, but this apparent inhibitory activity was an artifact caused by normalizing the firing rate of the cells to 3 seconds of baseline rather than 6 seconds as we have used for the other analyses. The wrong normalization created this artificial effect that looks like an inhibition. However, there is no suppression or entrained responses before the onset of the cues because the cues were presented at random intervals (30 to 90 s) and the animals were not able to predict when they were coming. We have now generated the correct normalization (6 s) and fixed the heatmaps in Fig. 2.

3. It is not discussed why VMH-PVT projection inactivation (Figure 8h-m) suppresses defensive behaviors, but doesn't increase food approach or lever pressing, while PVT or PVT-NAc inactivation (e.g. Figure 5b-g) does both. Could this be explained by the suppression of VMH collaterals to downstream brainstem targets that produce the defensive responses, rather than via PVT projections. This discrepancy needs to be discussed.

The reviewer is correct. It is possible that inhibition of VMH-aPVT neurons may also suppress collaterals of VMH to other regions, which could contribute to the observed reduction in defensive responses. However, our behavioral data in Supp. Fig. 6d argue against this possibility. We have added a new sentence in the discussion session to include this point:

“Furthermore, chemogenetic inactivation of VMHdm neurons attenuates freezing and risk-assessment behaviors in response to a predator¹, whereas photoactivation of these neurons induces a range of defensive behaviors that varies from avoidance, to freezing, to escape responses according to the intensity of laser stimulation² or the different VMHdm efferents activated³. Thus, it is possible that inactivation of collaterals from VMH-aPVT neurons to these two regions could mediate part of the reduced defensive responses observed in our study. However, our behavioral data showing that inhibition of VMH-aPVT neurons did not change rats' defensive behavior when the animals are exposed to cat odor alone suggest that such a possibility is minimal”.

To further explore the role of VMH-aPVT neurons in food seeking and defensive responses, we performed a battery of new experiments to photoactivate VMH-aPVT projections during the cued food-seeking test, the real time place preference test, and a new shuttle food-seeking test in which rats had to alternate between the two sides of the chamber to obtain food only in the side signaled by the food cues (See new Supp. Fig. 8). These experiments demonstrated that photoactivation of VMH-aPVT projections is sufficient to both attenuate food-seeking responses and induce avoidance responses in naïve animals. Thus, the lack of effects on food-seeking responses observed with inactivation of VMH-aPVT neurons is most likely due to the fact that, in addition to VMH, other inputs to aPVT may be also contributing to the

suppression in food-seeking responses observed during conflict. We have included the new findings in the results section and added a short paragraph in the discussion section to clarify this point:

“Our observation that inhibition of VMH-aPVT neurons reduced defensive behaviors but had no effect on food-seeking responses indicates that, in addition to VMH, other aPVT inputs may also contribute to the food-seeking suppression observed during conflict.”

4. Given the design of the box where the odor is right next to the food hopper any manipulation that increases defensive behaviors should also decrease level pressing for food – so how is it possible that a manipulation affects only defensive and not food responding? If you had the odor and food hopper at opposite ends of the cage, for example, would they still interfere? In other words, a circuit manipulation that decreases defense would appear to have the same effect as a manipulation that increases food seeking. How is this confound addressed in the study?

Thanks for the comments. This is indeed an interesting question that we are now addressing in our lab by using a modified version of our conflict paradigm in which animals have the chance to obtain food in both sides of the chamber (hidden and approach areas) during the cat odor presentation. We have preliminary data showing that animals exhibit a significant suppression in food-seeking responses even when the food dish is located in the hidden area far away from the cat odor source. While this is a completely different project in our lab, we have shared the results below for the reviewer’s appreciation.

In our study, we observed that two different manipulations (intra-aPVT infusion of Muscimol and chemogenetic inhibition of VMH-aPVT neurons) reduced defensive responses without

modifying food-seeking behaviors during the conflict test. Because distinct subtypes of neurons in the aPVT may regulate food-seeking in opposite directions, we believe that the lack of effect on food-seeking responses with pharmacological inactivation of aPVT may be due to the fact that Muscimol inactivated aPVT neurons in a non-selective way. In contrast, the lack of effect on food-seeking responses after VMH-aPVT inactivation could be explained by other aPVT inputs that may be also contributing to the suppression in food-seeking responses during conflict.

These two points were now clarified in the discussion session by adding the following sentences:

“Thus, distinct subtypes of neurons projecting from the aPVT to the NAc may modulate food-seeking responses in opposite directions. aPVT^{CRF} and Glut2-containing PVT neurons may work in synchrony to regulate the opposing pressures of predation and starvation. This functional heterogeneity could help to explain why our non-selective pharmacological inactivation of aPVT neurons had no effect in food-seeking responses”.

“Our observation that inhibition of VMH-aPVT neurons reduced defensive behaviors but had no effect on food-seeking responses suggests that, in addition to VMH, other aPVT inputs may also contribute to the food-seeking suppression observed during conflict”.

Reviewer #2 (Remarks to the Author):

I have to say that it has been a pleasure to read and evaluate this study, and I only have minor suggestions. My first suggestion is to tone down the conclusion about the functional role of the divergent connectivity between the MSN and CIN neurons of the NAc. Indeed, the authors did not record the synaptic strength or plasticity of this connection. I therefore suggest to change or remove “by mediating target-dependent synaptic transmission in the NAc” in the abstract and the result/discussion section.

We agree and have removed the sentence from the abstract and toned down the interpretation for the remainder of the manuscript.

As the main techniques is in vivo electrophysiological recordings, and although I trust recordings quality, showing examples raw traces of multiple channels, and some clusters in Figure 2a or in a supplementary figure would be beneficial.

We have incorporated some examples of raw data showing waveforms and spike clusters with the principal component analyses of 3 representative channels. Please see Supplementary Figure 5.

To avoid misunderstanding please replace aPVT-projecting VMH neurons with VMH-aPVT neurons.

We have followed the reviewer’s recommendation and replaced these terminologies. We have changed aPVT-projecting VMH neurons to VMH-aPVT neurons, as well as NAc-projecting aPVT neurons to aPVT-NAc neurons.

The authors provide a large number of long videos. To increase the value of this supplementary files, I would advise the authors to show all of them at 16x speed all the time rather than changing speed in the middle of videos and at different speeds across videos. Also, it would greatly increase the impact of the video to show them simultaneously rather than one after the other: mCherry on the top, DREADDs on the bottom – same for ChR/eYFP etc. Also, please keep the experimental group of the rat the whole time on the video. Finally, this is optional, but a great addition would be to add the track of the animal on the video in red.

We appreciate the reviewer's recommendation. We have now rearranged the videos to show the control and experimental groups simultaneously. We have also modified the legends accordingly. In Supplementary Videos 1 and 2, we decided to keep the normal speed of the videos to make sure that readers can appreciate a series of defensive responses that are exhibited by the rats during the cat odor exposure and conflict tests.

Figure 1

- k-o: please specify the y axis titles : is it raw numbers of cells ? or cells/mm² (which it should be)

Thanks for catching this. We have fixed the Y axis label to cells/mm².

- b: the color code seems to be black for neutral odor and green for cat odor – however, these are mixed up in c-h and k-o. Please make the neutral odor histograms black, as well as the dots, for the cat odor, keep the histograms green (ideally, filling 50% transparent and outline 100% green) and the dots with 100% green outline and white filling.

Also, in a, the neutral odor is grey – please make it black

- b and j: rather than rectangles surrounding the data, the authors could consider a thick black or green line on top of the graphs, and right below 'Neutral odor' and 'Cat odor'.

- same comments apply to Figure S1 and all figures.

We appreciate the reviewer for looking carefully to each one of our figures. We have accepted the reviewer's recommendation to change the color of the figures for the control groups. We believe that the figures look much better now.

Figure 2

- it remains unclear why the authors do not represent the same analysis for the cat odor than for the food-seeking (neurons responding to cat odor before vs. same neurons during conflict, as well as neurons responding during conflict vs. same neurons before conflict). I understand the time locked response with the odor is more difficult to define, however, they characterized 11% of the recorded neurons as 'cat odor responsive'. Their response during conflict should be assessed, at least in a supplementary figure

We apologize for any misunderstanding that Fig. 2c may have caused. The pie chart in this figure shows the changes in spontaneous firing rate of the cells after each phase of the test. The 11% of cat odor responsive cells refer to neurons that modified their firing rate exclusively after the 10 min of cat odor exposure. As the reviewer mentioned, time-locking the firing rate of the cells to each time the animals smelled the cat odor may result in a very inaccurate alignment and in many cases a reduced number of events because the animals will avoid the cat odor source. Therefore, to try to address the reviewer's recommendation, we have extended our analyses by comparing changes in spontaneous firing rate between the cat odor and the conflict phases (please see Supplementary Fig. 4h).

- c: It is written in the legend, but adding a legend on the figure that red is excitation and blue inhibition would be helpful.

We have added a legend for excitation and inhibition in Figure 2c as recommended.

Figure 2c/4j: why did the authors choose different representations for the proportion of neurons? Please be consistent. Moreover, statistical comparisons of these proportions would be very interesting.

We have followed the reviewer's recommendation and replaced the stack bar graphics in Fig. 4J by pie charts to match Fig. 2c. In addition, we have added a new analysis showing a breakdown with the proportions of the non-selective responses in Supp. Figure 4h-i.

Figure 8: part of the text has a different font than the rest of the figure.

We thank the reviewer again for catching this mistake. We have fixed it.

In the discussion, the authors could consider citing the other electrophysiological studies during conflict of positive and negative cues such as Burgos-Robles et al. 2017 (PMID: 28436980).

We have now cited the article in the discussion session.

Reviewer #3 (Remarks to the Author):

While I do not consider that further experimental work will be strictly necessary, below I raised several points that require clarification from the authors:

1) The observation that inhibition of the aPVT doesn't abolish defensive responses to 'Cat odor only' is very interesting. This observation seems to be in line with the notion that the PVT is not required for acute fear responses (for example during fear conditioning) but rather mediates fear responses to conditioned stimuli (fear memory retrieval). However, defensive responses during conflict are dependent on the aPVT. The authors should offer more conceptual insight into why they think that is. Such insight could help to advance important questions of the field.

We appreciate the reviewer for raising this point. We believe that in aPVT, CRF neurons play a critical role in situations of motivational conflict when animals need to use environmental cues to regulate their approach-avoidance responses. This is supported by our recording findings in Fig. 4 showing that aPVT^{CRF} neurons are preferentially recruited during the conflict test. The convergence of inputs conveying food- and predator-related information to aPVT^{CRF} neurons seem to be the critical point to recruit these cells. In fact, all aPVT inactivation we performed in this study altered rat's behavior during the conflict, but had no effect on food-seeking or defensive responses when the two tasks were performed separately. Therefore, in the absence of conflict, when a smaller fraction of aPVT^{CRF} cells is recruited, aPVT inactivation doesn't reveal any behavioral effect. We have discussed the preferential role of PVT^{CRF} neurons during conflicting situations at different parts of the discussion:

PAGE 18 - *"Therefore, in risky environments, animals are expected to fine-tune their gradient of defensive responses to enable foraging behavior². These considerations suggest the*

existence of a common node that integrates anti-predator defenses and food-related information to regulate the competing drives of approaching food and avoiding potential threats. Our results showing that aPVT neurons change their firing rate in response to both food cues and predator odor indicate that this thalamic region is a potential candidate for this integrative role”.

PAGE 18 – “The lack of effect of aPVT inactivation on innate defensive responses in the absence of conflict seems at odds with previous studies showing that inactivation of PVT impairs the retrieval of conditioned fear responses 8, 9. However, this apparent discrepancy may be attributed to differences in the neural circuits that mediate anti-predator and conditioned defensive responses 42, or alternatively, differences in the target of the inactivation between the current study (aPVT) and the previous reports (pPVT), as these two subregions of PVT exhibit genetically distinct subtypes of cells and neuroanatomical connections 17, 21, 43.”

PAGE 19 – “Our observation that aPVT^{CRF} neurons are activated by cat odor, but indispensable for food-seeking suppression exclusively during conflict, argues in favor of a critical role of these cells in regulating behaviors with opposite motivational drives. In support of this idea, we found that aPVT^{CRF} neurons send dense glutamatergic projections to the NAcSh, a region implicated in reward-seeking motivation 22, 23, and photoactivation of the aPVT^{CRF}-NAc pathway in a neutral environment suppresses food-seeking and elicits avoidance responses in a way that resembles the effects of cat odor exposure”

2) Using cFos, the authors conclude that aPVT^{CRF} neurons are activated during the conflict (cat odor). Interestingly, while in photo-tagged aPVT^{CRF} neurons they also observed an increase in the percentage of neurons modulated by the food cue during the conflict compared to before the conflict, the more prominent effect observed here is in neurons that are inhibited by the food cue. This particular finding seems to be at odds with the authors' conclusions about the role of CRF neurons in the PVT. Can the authors clarify this issue?

We appreciate the reviewer for the thoughtful question. We believe that the aPVT^{CRF} neurons that are activated by food cues during the conflict phase project mainly to the NAc (anti food seeking), whereas the PVT-CRF neurons that are silenced during conflict send projections to other brain regions (pro food seeking). We have clarified this point in the discussion section: *“Our observation that aPVT^{CRF} neurons exhibit either excitatory or inhibitory food-cue responses during conflict suggests the existence of distinct subtypes of aPVT^{CRF} neurons, which may have different anatomical targets. We speculate that the excitatory food-cue responses recorded during conflict arise from aPVT^{CRF}-NAc neurons, which provide a stop-signal to accumbal neurons to suppress food seeking in the presence of potential threats. A recent study has shown that aPVT-NAc neurons receive inputs from PL and LH neurons, which respectively contribute for cued-food seeking and feeding responses 4. We have found that both PL and LH neurons send direct projections to aPVT^{CRF} neurons, but how these pathways contribute for the balance between approaching food and avoiding predator threats will be the subject of further studies”.*

3) The observation that unlike whole aPVT silencing, exclusively silencing aPVT-NAc neurons rescues food pressing behavior during the conflict suggests that PVT circuits controlling food and aversive-related behaviors are somewhat segregated. The authors should offer additional insight on this point.

We agree with the reviewer that different populations of aPVT neurons may regulate food seeking and defensive behaviors in opposite directions and they may be partially segregated. This would explain why non-selective inactivation of aPVT neurons during conflict had no

effect on food seeking when compared to cell-type or projection-defined manipulations. Our single-unit recordings suggest that the same population of aPVT^{CRF} neurons are able to encode information about both food cues and cat odor, thereby suggesting that aPVT^{CRF} neurons may integrate food signals with predator information to regulate approach-avoidance responses during conflict.

We have expanded our discussion now to clarify this point: *“Thus, distinct subtypes of neurons projecting from the aPVT to the NAc may modulate food-seeking responses in opposite directions. aPVT^{CRF} and Glut2-containing PVT neurons may work in synchrony to regulate the opposing pressures of predation and starvation. This functional heterogeneity could help to explain why our non-selective pharmacological inactivation of aPVT neurons had no effect in food-seeking responses.”*

4) Related to the previous point, the authors identify the VMH as providing critical input to aPVT^{CRF} neurons during conflict. However, whereas silencing aPVT^{CRF} neurons diminishes defensive behaviors and rescues food seeking, silencing of aPVT-projecting VMH neurons attenuates defensive behaviors but do not rescue food seeking. This result is consistent with segregation of the aPVT circuits that promote defensive behavior and suppress reward seeking during conflict. There is a sense that heterogeneous populations of PVT neurons guide behavior during conflict. The authors should address these seemingly discrepant observation.

We agree with the reviewer that aPVT contains segregated populations of neurons that respond exclusively to food seeking and defensive behaviors, as we have demonstrated in Fig. 2c. However, in the same figure we have also found that most of the responsive cells in the aPVT modified their firing rates in response to either food seeking or cat odor stimuli. We have made some additional analyses to demonstrate that 51% of non-selective aPVT neurons that change their activity during food seeking respond in opposite directions during cat odor (i.e., valence coding, see Supplementary Fig 4h), whereas 71% of non-selective aPVT neurons that change their activity during cat odor respond in opposite directions during conflict (See Supplementary Figure 4i) This observation reinforces the idea that a specific population of aPVT neurons can integrate rewarding and aversive information during motivational conflict.

To further explore the lack of effect in food-seeking responses following inhibition of VMH-aPVT neurons during conflict, we performed a battery of new experiments to photoactivate VMH-aPVT projections during the cued food-seeking test, the real time place preference test, and a new shuttle food-seeking test in which rats had to alternate between the two sides of the chamber to obtain food only in the side signaled by the food cues (See new Supp. Fig. 8). These experiments demonstrated that photoactivation of VMH-aPVT projections is sufficient to both attenuate food seeking and induce avoidance responses in naïve animals. Thus, the lack of effects on food-seeking responses observed with inactivation of VMH-aPVT neurons is most likely due to the fact that, in addition to VMH, other aPVT inputs may be also contributing to the suppression in food-seeking responses observed during conflict. We have included the new findings in the results section and added a short paragraph in the discussion section to clarify this point:

“Our observation that inhibition of VMH-aPVT neurons reduced defensive behaviors but had no effect on food-seeking responses indicates that, in addition to VMH, other aPVT inputs may also contribute to the food-seeking suppression observed during conflict.”

5) Also, the authors show that stimulation of aPVTCRF neurons mimics the defensive (avoidance) responses and attenuated food seeking observed during conflict. But what about freezing behavior? Inactivation of aPVT-NAc neurons attenuates freezing (Figure 3). Thank you for raising this important question. We have now analyzed the freezing responses induced by photoactivation of aPVT^{CRF} neurons as well as aPVT^{CRF}-NAc projections during both the cued food-seeking and the RTPP tests. We observed that photoactivation of aPVT^{CRF} neurons or aPVT^{CRF}-NAc projections had no effect on the expression of freezing behavior (See Fig. 5i-j and Fig. 7b-c). These results suggest that while aPVT neurons and their projections to the NAc seem to be necessary to regulate freezing responses during a conflicting situation, their activation is not sufficient to induce freezing responses in a neutral context.

REFERENCES

1. Silva, B.A., et al. Independent hypothalamic circuits for social and predator fear. *Nat Neurosci* 16, 1731-1733 (2013).
2. Kunwar, P.S., et al. Ventromedial hypothalamic neurons control a defensive emotion state. *Elife* 4 (2015).
3. Wang, L., Chen, I.Z. & Lin, D. Collateral pathways from the ventromedial hypothalamus mediate defensive behaviors. *Neuron* 85, 1344-1358 (2015).
4. Otis, J.M., et al. Paraventricular Thalamus Projection Neurons Integrate Cortical and Hypothalamic Signals for Cue-Reward Processing. *Neuron* 103, 423-431 e424 (2019).

Reviewers' Comments:

Reviewer #1:

Remarks to the Author:

I am satisfied with the authors' response to my criticism and comments.

Reviewer #2:

Remarks to the Author:

The authors have address all my comments and I believe the study is now ready for publication. The manuscript, figures and videos have greatly improved, and I will only make three small suggestions to slightly improve them even more.

1. The authors do not provide controls for the behavioral impact of the chemo-genetic and opto-genetic manipulations. If the authors performed these manipulations during an open field test, analyzing the locomotor activity (distance travelled and speed) would be a great addition to strengthen the specificity of their finding.

2. in Figure 1 c-h the color code is not fully accurate. Indeed during baseline, the odor presented is a neutral odor, but the legend for green is 'cat odor'. The legend should be 'cat odor group'. To make things clearer the authors should change the legends to 'neutral odor group' for grey and 'cat odor group' for green, and the 2 groups should be defined in Figure 1a:
'Neutral odor group': neutral odor during baseline and test
'Cat odor group': neutral odor during baseline and cat odor during test
This comment also applies to figures 3, 5 and S3.

3. The presentation of the videos have greatly improved, but I would advise that the name of the neural populations manipulated remain visible during the entire video, not only briefly at the beginning.

Reviewer #3:

Remarks to the Author:

In their revised manuscript and accompanying rebuttal letter, Engelke and colleagues have addressed my previous concerns in full. I have no further comments.

Responses to Reviewer 2

We thank Reviewer 2 for the additional recommendations. Please find below a point-by-point response to the comments.

Reviewer #2 (Remarks to the Author):

The authors have address all my comments and I believe the study is now ready for publication. The manuscript, figures and videos have greatly improved, and I will only make three small suggestions to slightly improve them even more.

1. The authors do not provide controls for the behavioral impact of the chemo-genetic and opto-genetic manipulations. If the authors performed these manipulations during an open field test, analyzing the locomotor activity (distance travelled and speed) would be a great addition to strengthen the specificity of their finding.

Response: We have now analyzed the total distance travelled and maximum speed for the experimental groups and the control animals for both optogenetic and chemogenetic manipulations involving aPVT^{CRF} neurons. We haven't observed any significant effects of either the chemogenetic or the optogenetic manipulations on general locomotor activity. We have included new graphics in Fig 5q-r, 7j-K, and Supp Fig 6g to display the results.

2. in Figure 1 c-h the color code is not fully accurate. Indeed during baseline, the odor presented is a neutral odor, but the legend for green is 'cat odor'. The legend should be 'cat odor group'.

To make things clearer the authors should change the legends to 'neutral odor group' for grey and 'cat odor group' for green, and the 2 groups should be defined in Figure 1a:

'Neutral odor group': neutral odor during baseline and test

'Cat odor group': neutral odor during baseline and cat odor during test

This comment also applies to figures 3, 5 and S3.

Response: We have now made all the recommended alterations to clarify the experimental conditions for each one of the groups as suggested by the reviewer.

3. The presentation of the videos have greatly improved, but I would advise that the name of the neural populations manipulated remain visible during the entire video, not only briefly at the beginning.

Response: Good point. We have modified that accordingly.